# Probing strigolactone perception mechanisms with rationally designed small-molecule agonists stimulating germination of root parasitic weeds

Dawei Wang[1,5], Zhili Pang[1,5], Haiyang Yu[2], Benjamin Thiombiano[3], Aimee Walmsley[3], Shuyi Yu[1], Yingying Zhang[1], Tao Wei[1], Lu Liang[1], Jing Wang[4], Xin Wen[1], Harro J. Bouwmeester [3✉], Ruifeng Yao [2✉] & Zhen Xi [1✉]

The development of potent strigolactone (SL) agonists as suicidal germination inducers could be a useful strategy for controlling root parasitic weeds, but uncertainty about the SL perception mechanism impedes real progress. Here we describe small-molecule agonists that efficiently stimulate *Phelipanchce aegyptiaca*, and *Striga hermonthica*, germination in concentrations as low as $10^{-8}$ to $10^{-17}$ M. We show that full efficiency of synthetic SL agonists in triggering signaling through the *Striga* SL receptor, ShHTL7, depends on the receptor-catalyzed hydrolytic reaction of the agonists. Additionally, we reveal that the stereochemistry of synthetic SL analogs affects the hydrolytic ability of ShHTL7 by influencing the probability of the privileged conformations of ShHTL7. Importantly, an alternative ShHTL7-mediated hydrolysis mechanism, proceeding via nucleophilic attack of the NE2 atom of H246 to the 2′C of the D-ring, is reported. Together, our findings provide insight into SL hydrolysis and structure-perception mechanisms, and potent suicide germination stimulants, which would contribute to the elimination of the noxious parasitic weeds.

[1] State Key Laboratory of Elemento-Organic Chemistry and Department of Chemical Biology, National Pesticide Engineering Research Center, Collaborative Innovation Center of Chemical Science and Engineering, College of Chemistry, Nankai University, 300071 Tianjin, P. R. China. [2] State Key Laboratory of Chemo/Biosensing and Chemometrics, Hunan Provincial Key Laboratory of Plant Functional Genomics and Developmental Regulation, College of Biology, Hunan University, 410082 Changsha, P. R. China. [3] Swammerdam Institute for Life Sciences (SILS), University of Amsterdam, Science Park 904, 1098 XH Amsterdam, Netherlands. [4] State Key Laboratory of Natural and Biomimetic Drugs, School of Pharmaceutical Sciences, Peking University, 100871 Beijing, P. R. China. [5] These authors contributed equally: Dawei Wang, Zhili Pang. ✉email: H.J.Bouwmeester@uva.nl; ryao@hnu.edu.cn; zhenxi@nankai.edu.cn

Root parasitic weeds, such as the witchweeds, *Striga* spp., and broomrapes, *Orobanche* and *Phelipanche* spp., rely on a host plant to obtain nutrients and water[1,2] and severely affect crop production worldwide[3]. *Striga* has a wide distribution in sub-Saharan Africa, and mainly infests cereals, causing annual losses of more than 10 billion US dollars[4]. *Orobanche* is widely distributed in arid and semi-arid areas, and mainly infests non-cereals, leading to billions of dollars in crop losses every year. For example, in China's Xinjiang province, broomrape *Phelipanche aegyptiaca* and *Orobanche cumana* Wallr have heavily affected more than 60,000 hectares of arable lands[5]. Single plants of these parasitic weeds can produce up to hundreds of thousands of tiny seeds. The seeds get buried in the soil and remain viable for up to 20 years[6]. This results in the establishment of a seed bank of the parasitic weeds in infested soils, which poses a great challenge in combating them. The germination of the seeds of most of these parasitic weeds requires strigolactones (SLs), a class of apocarotenoid compounds present in the root exudates of plants[7–9]. After germination the parasitic plants cannot survive without a host and, therefore, the use of seed germination stimulants to induce suicidal germination and hence eradicate the seed bank of these obligate parasites may be a good strategy to control parasitic weeds in agriculture.

SLs have a key role not only in inducing parasitic weed germination but also act as endogenous phytohormones that regulate plant development, such as shoot branching and root development[10–14]. SLs all have a conserved enol-ether butenolide functional group with *R*-configured 2′C (called the D-ring) (Supplementary Fig. 1)[15–17]. A large number of studies demonstrated that the 2′*R* coupled D-ring is essential for the biological functions of the SLs[17,18]. Reducing the double bond in the D-ring but also substitution with a carbon of the oxygen in the enol-ether bridge significantly impairs bioactivity[19].

A family of related α/β-hydrolase proteins has been suggested to play a role in SL perception. For example, DWARF14 (D14) proteins have been identified as the SL receptor that inhibits shoot branching[20,21]. HYPOSENSITIVE TO LIGHT/KARRIKIN INSENSITIVE2 (HTL/KAI2) proteins have been reported to be the receptors that regulate seed germination in parasites in response to SLs[22–24]. Intriguingly, HTL/KAI2 proteins have undergone convergent evolution allowing for SL perception, just as D14[25]. All these SL receptors have a conserved Ser-Asp-His catalytic triad at the bottom of their binding pockets. SL receptors are noncanonical hormone receptors as they irreversibly bind SLs and generate a covalently linked intermediate molecule (CLIM, $C_5H_5O_2$) connected to the histidine of the catalytic triad[23,26–28], as judged from the CLIM–AtD14–D3 (the F-box component DWARF3)–ASK1 (*Arabidopsis* SKP1-like1; SKP1, S-phase kinase-associated protein 1) crystal complex[27]. However, after reanalyzing the electron density maps of the crystal complex, Carlsson et al. believed that an iodine ion would fit better than CLIM to the density inside the crystal AtD14 structure[29], while Bürger et al., refined the X-ray data and proposed that the D-ring would fit well in the histidine-butenolide complex and emphasized the necessity of covalent modification by the D-ring[15]. Besides, Seto et al. found that AtD14[D218A] could bind, but failed to hydrolyze SLs, while SLs still enhanced AtD14[D218A]−SMXL7 interaction in vitro and AtD14[D218A] restored the highly branched phenotype of the *atd14 max4* double mutant in an exogenously applied SL-dependent manner. Therefore, the authors proposed that the active signaling state of D14 was initiated by intact SLs, rather than covalent modification by the D-ring[15]. Shabek et al. reported the existence of two-variable structural states of the C-terminal α-helix (CTH) domain of D3. The dislodged CTH promotes D3 to bind the open conformational D14 and prevents D14 from hydrolyzing SLs, while the engaged form could interact with CLIM-bound closed D14[30]. Although SLs can induce signal propagation via the SL receptor-MAX2 pathway, the perception mechanism remains a subject of debate[15,29,31].

In this study, we report the discovery of aryloxyacetyl piperazines as potent *Phelipanchce aegyptiaca* and *Striga hermonthica* seed germination stimulants by using hierarchical virtual screening and structural optimization approaches. Among them, compound (*S*)-**4a** showed excellent germination activity outperforming that of *rac*-GR24, one of the most active and widely used synthetic SL analogs. Using molecular dynamics (MD) simulation and statistical analysis, we found that the stereochemistry of synthetic SL analogs affected the hydrolytic efficacy of ShHTL7 by inducing ShHTL7 to preferentially acquire a higher probability of the privileged conformation. Finally, the involvement of a ShHTL7-mediated hydrolysis mode of the synthetic agonists in activating the signaling state is proposed. Through the rational design of small molecular probes, our work provides insight into SL perception and SL-related chemical biology.

## Results

**Identification of small molecules binding to ShHTL7 with micromolar affinity**. Among the reported SL receptors, ShHTL7 was identified to show hypersensitivity to various types of SL agonists, which could potentially be attributed to the relatively large ligand-binding pocket of the protein[32,33]. To search for compounds binding to ShHTL7, we adopted a hierarchical 2D molecular similarity-based virtual screening using 13 reported parasitic plant seed germination stimulants as bait (Fig. 1a, Supplementary Fig. 1)[34]. A chemical library containing more than 210,000 compounds from Specs database (www.specs.net) was screened using the Find Similar Molecules module in the Accelrys Discovery Studio 2.5 (DS 2.5) following the Fingerprints protocol[35]. The top 100 compounds were then clustered based on the fingerprints, and 10 compounds in the cluster centers were selected. This selection was expanded to a total of 51 candidates by further selection from the top 130 ranked compounds based on structural features (Supplementary Fig. 2).

We used a sensitive, high-throughput Surface plasmon resonance (SPR) method to examine the binding affinities of the 51 candidates to the recombinant ShHTL7 protein, which was immobilized on the CM5 sensor chip. *rac*-GR24 was used as a positive control. Using kinetics analysis in the Biacore evaluation software (Biacore T200, Version 2.0) the binding affinity ($K_D$) value of *rac*-GR24 to ShHTL7 was determined to be about 0.93 μM, similar to the previously published result ($K_d = 0.92 \pm 0.01$ μM in microscale thermophoresis measurements)[2]. Among the tested candidates at 50 μM, (4-(cyclohex-3-en-1-ylmethyl)piperazin-1-yl)(4-fluorophenyl)methanone, designated **1**, had the highest affinity to the recombinant ShHTL7. The $K_D$ value of **1** was $13.3 \pm 1.4$ μM. Several compounds with lower affinity were also found but not studied further (Supplementary Fig. 3).

Having identified a candidate ligand that can bind ShHTL7 at the micromole level, we then performed the second round of screening by using **1** as the reference molecule. Twenty-seven candidates with different substituents on both sides of the piperazine ring of **1** were selected for further binding affinity assays (Supplementary Fig. 4). In the second round of SPR, we examined the $K_D$ values of compounds that showed higher binding levels (at 50 μM) than compound **1**. Eight hits, designated **2a-h**, with improved binding affinity to ShHTL7, compared with **1**, were found (Fig. 1b). Among these eight hits, compound **2h** bound to ShHTL7 with a $K_D$ of 0.96 μM (Fig. 1c), the highest affinity to ShHTL7 detected in the current study. To find a suitable scaffold for further optimization, we investigated the preliminary structure-activity relationships of these compounds. Compounds with bicyclo[2.2.1]hept-5-en-2-yl

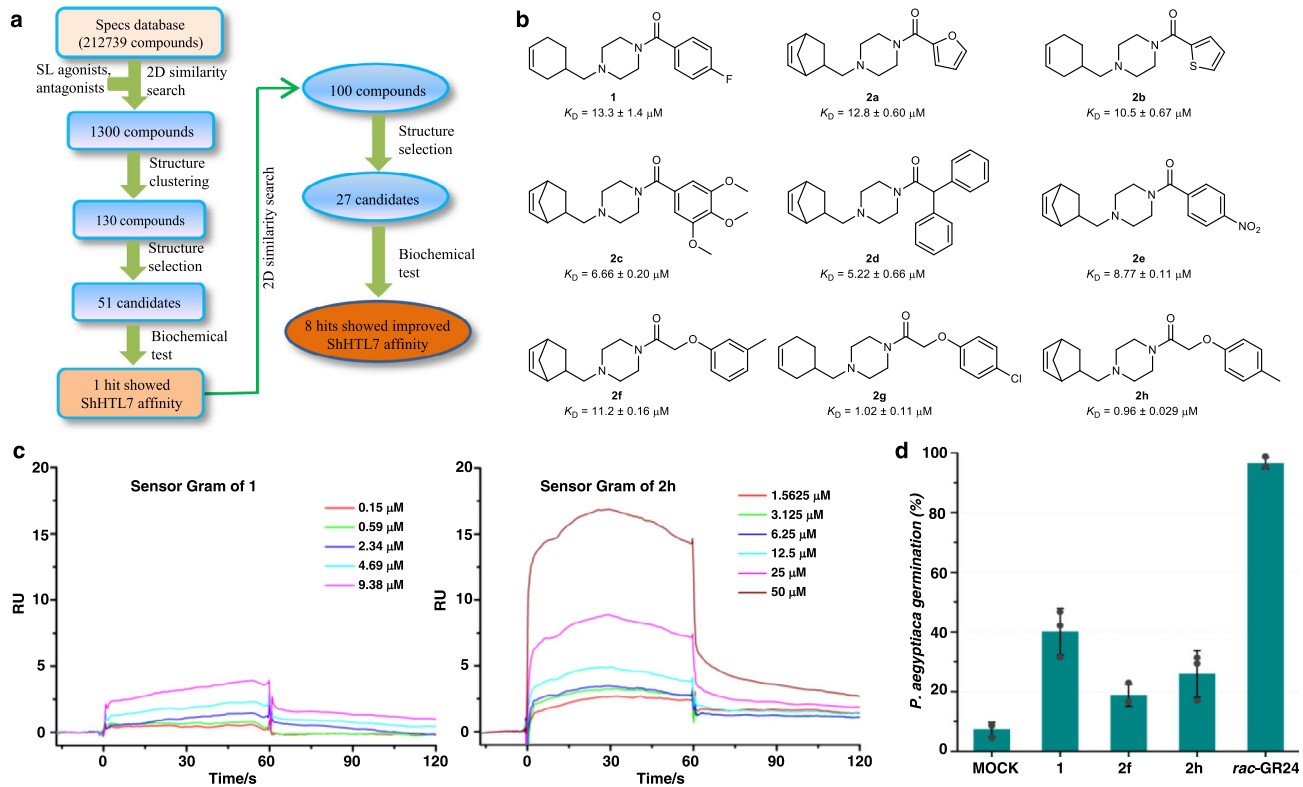

**Fig. 1 The discovery of small molecules binding to ShHTL7 by virtual-screening approach. a** A hierarchical 2D molecular fingerprints similarity-based virtual-screening strategy was performed: a database containing more than 210,000 compounds from Specs was screened by Accelrys Discovery Studio 2.5 (DS 2.5). Thirteen reported parasitic plant seed germination stimulants (Supplementary Fig. 1) were selected as representative search query molecules, 78 candidates were selected for ShHTL7 binding assay, which depended on their physicochemical properties and structural features, with nine hits found. **b** Chemical structures of nine hit compounds, the $K_D$ values were determined by SPR assay and the results are shown below each hit. **c** Representative figures of SPR sensorgrams of compounds **1** and **2h** from three independent experiments. **d** Compounds **1**, **2f**, **2h**, and *rac*-GR24 induced the germination of *P. aegyptiaca* seeds at 10 μM. Error bars mean standard error (SD, $n = 3$ biologically independent experiments).

groups generally showed higher affinities than cyclohex-3-en-1-yl substituted analogs. For the heterocyclic-substituted compounds, five-membered heterocycles showed higher affinity than six-membered heterocycles (Fig. 1b, Supplementary Fig. 4). Substituent positioning *para* with respect to the benzoyl groups (e.g., **1**) was favorable for binding with ShHTL7 (e.g., **2f** vs **2h**). We also observed that compounds with electron-withdrawing groups (e.g., **2e**, Fig. 1b) on the benzoyl moiety had a smaller $K_D$ than those with electron-donating groups (e.g., **2i**, Supplementary Fig. 4).

To test whether these candidate HTL ligands have seed germination stimulation ability in root parasitic weeds, we selected *P. aegyptiaca*, one of the most destructive root parasitic weeds in the world, to perform bioactivity tests. Compounds **1**, **2f**, and **2h** stimulated the germination of *P. aegyptiaca* in a concentration as low as 10 μM, (Fig. 1d, Supplementary Fig. 5). Considering the promising results with SPR in the *P. aegyptiaca* germination assay, we decided to pursue further structural optimization of the lead compound **2h**.

**Structural optimization of aryloxyacetyl piperazines to create highly potent *P. aegyptiaca* and *S. hermonthica* seed germination agonists.** Based on the structural feature of the lead compound **2h**, we selected three regions to perform further modification (Supplementary Fig. 6), to try to further improve bioactivity. Insertion of a carbonyl group between the bicyclo[2.2.1]hept-5-en-2-yl group and the piperazine ring was detrimental to binding affinity (Supplementary Fig. 6, region A). Removal of an oxygen atom in region B was found unfavorable to binding affinity as well. Therefore, we focused on modifying the benzene ring (region C) for

**2h**. A series of compounds with different groups at region C was synthesized (Supplementary Scheme 1). Initial SPR screening results indicated that among the tested compounds, four compounds (**3d-g**) showed equal or higher binding affinity compared with that of **2h**. The $K_D$ values of **3d**, **3e**, **3f**, and **3g** were 1.86, 0.65, 0.18, and 0.47 μM, respectively. The separation of **3f** into pure stereoisomers showed that (2*S*)-**3f** displayed a remarkably improved binding affinity ($K_D = 1.28 ± 0.13$ nM) compared with **2h** ($K_D = 0.96 ± 0.029$ μM) and (2*R*)-**3f** ($K_D = 0.48 ± 0.011$ μM). However, when testing the *P. aegyptiaca* seed germination-inducing activity of the compounds **3a-r**, no significantly improved seed germination activity was observed compared to that of **2h** at the concentration of 10 μM (Supplementary Table 1). As SPR just measures the binding ability, another independent assay by using Yoshimulactone Green (YLG) as the fluorescent substrate was performed to determine the half-maximal inhibitory concentration ($IC_{50}$) of these compounds against ShHTL7[22]. The results show that the compounds display competitive inhibition of ShHTL7 activity at micromolar concentrations (Supplementary Fig. 7, Supplementary Table 1). The ineffective competitive inhibition of ShHTL7 and lack of improved seed germination-inducing activity of these candidates may suggest that ShHTL7-mediated hydrolysis plays an important role in inducing seed germination[23].

Many studies have shown that the D-ring plays a critical role in the modulation of SL activity[15,36]. For example, Samejima et al. reported a highly active carbamate-D-ring containing compound T-101[37]; Uraguchi et al. also introduced the carbamate-D-ring scaffold into their structure and obtained a highly active compound SPL7[28]. We, therefore, inferred that introducing the

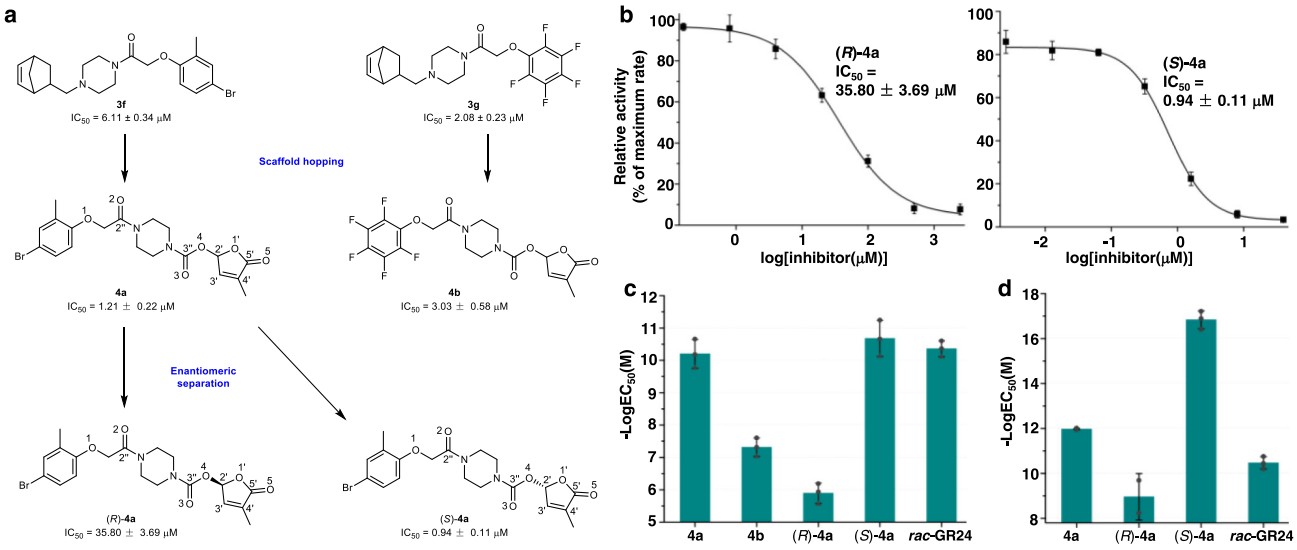

**Fig. 2 Rational optimization yields the discovery of highly potent seed germination stimulants for *P. aegyptiaca* and *S. hermonthica*. a** Structural optimization process of compounds **3f** and **3g**. **b** Competitive ShHTL7 inhibitory activity curve via YLG assay under different concentrations of (*R*)-**4a** and (*S*)-**4a**, error bars mean SD ($n = 3$ biologically independent experiments). **c** The *P. aegyptiaca* germination $EC_{50}$ values of compounds **4a**, **4b**, (*R*)-**4a**, (*S*)-**4a**, and *rac*-GR24, error bars mean SD ($n = 3$ biologically independent experiments). **d** The *S. hermonthica* germination $EC_{50}$ values of compounds **4a**, (*R*)-**4a**, (*S*)-**4a**, and *rac*-GR24, error bars mean SD ($n = 2$ or 3 biologically independent experiments).

carbamate-D-ring to the structure of aryloxyacetyl piperazines might improve the seed germination bioactivity of *P. aegyptiaca* as well. The carbamate-D-ring moiety was introduced into the aryloxyacetyl piperazines of **3f** and **3g**, respectively, by adopting the "scaffold hopping" strategy[38]. Compounds **4a** and **4b** were then successfully synthesized (Supplementary Scheme 2). As illustrated in Fig. 2a, compared with their corresponding parent compounds, **4a** had a lower $IC_{50}$ value than **3f**, whereas **4b** displayed an increased $IC_{50}$ value compared with **3g**. Although the $IC_{50}$ values in the YLG assay differed, both **4a** and **4b** showed significantly improved seed germination-inducing activity in *P. aegyptiaca* when compared with that of **3f** and **3g**, respectively. The YLG assay was performed with ShHTL7, not the protein homologs from *P. aegyptiaca*; so it is not surprising that the results do not completely match. The half-maximum effective concentrations ($EC_{50}$) of **4a** and **4b** were determined around $8.62 \times 10^{-11}$ M and $5.52 \times 10^{-8}$ M, respectively (Fig. 2c). Inspired by this, we then hybridized the carbamate-D-ring scaffold with the 4-(bicyclo[2.2.1]hept-5-en-2-ylmethyl)piperazin-1-yl group of **3f** and synthesized compound **4c**. **4c** showed increased $IC_{50}$ ($4.56 \pm 0.78$ μM) and $EC_{50}$ ($2.26 \times 10^{-9}$ M) values compared to that of **4a** (Supplementary Fig. 8). Notably, our data demonstrated that installing the carbamate-D-ring warhead into the target compounds can significantly improve the *P. aegyptiaca* seed germination potency. These effects can also be observed in compounds **3d** → **4d**, and **3e** → **4e** (Supplementary Fig. 8).

Based on the obtained lead compound **4a**, additional attempts were made to explore the effect of substituents changes on the bioactivity. Introducing two fluorine atoms at the methylene of the oxyacetyl group resulted in a substantial decrease in seed germination stimulating activity (**4f**, Supplementary Fig. 8). Any changes in the carbamate-D-ring scaffold, including replacing the 1′ oxygen atom of the D-ring with methylene (**4g**) or *N*-methyl (**4h**) group lead to a significant drop in seed germination activity (Supplementary Fig. 9), suggesting that the excellent germination activity of **4a** heavily relied on the integrity of the carbamate-D-ring. Additionally, extensive studies have demonstrated that the configuration at the 2′C of the D-ring is crucial to the bioactivity of SL analogs, and compounds with 2′R configuration showed better

bioactivity than their 2′S enantiomers[17,39]. To investigate the effect of the stereochemistry of **4a** on bioactivity, we isolated both enantiomers of **4a** by preparative chiral high-performance liquid chromatography (HPLC). As illustrated in Fig. 2b, c, (*S*)−**4a** displayed a more than 30-fold increase in the $IC_{50}$ value and over 40,000-fold improved *P. aegyptiaca* seed germination activity than (*R*)−**4a**. It is worth mentioning that (*S*)−**4a** even exhibited a slightly higher seed germination activity than *rac*-GR24, as evident from $EC_{50}$ values of (*S*)−**4a** and *rac*-GR24 at $3.39 \times 10^{-11}$ M and $4.85 \times 10^{-11}$ M, respectively. Inspired by the excellent *P. aegyptiaca* seed germination-inducing activity of (*S*)-**4a**, we furtherly analyzed the bioactivity against another damaging root parasitic weed *S. hermonthica*. Similarly to *P. aegyptiaca* (*S*)-**4a** exhibited slightly higher *S. hermonthica* seed germination activity in comparison to *rac*-GR24 (Fig. 2d). Taken together, our results show that the carbamate-D-ring and the configuration of **4a** are critical for HTL receptor binding and seed germination-inducing activity.

**Covalent modification of ShHTL7 enabled the excellent bioactivity**. Chemical reactivity is an important factor in the bioactivity of natural and synthetic SLs. Therefore, we evaluated the reactivity of compounds **4a**, (*R*)-**4a**, (*S*)-**4a**, **4g-i**, and *rac*-GR24 in a 1:3 mixture of methanol and phosphate-buffered saline (PBS, 10 mM) at pH 7.4 by HPLC. The half-life times ($t_{1/2}$) of **4a**, (*R*)-**4a**, and (*S*)-**4a** were determined to range from 3.4 to 7.7 h, a slightly lower reactivity than that of *rac*-GR24 ($t_{1/2} = 3.0$ h) (Supplementary Fig. 10). It should be noted that **4h** ($t_{1/2} = 1.1$ h) displayed higher reactivity when compared with that of **4a**. This is due to the higher electron-donating effect of *N*-CH$_3$ of the D-lactam functional group of **4h** compared to the oxygen of the D-ring of **4a**, facilitating the detachment of the D-lactam from the rest of the molecule. In marked contrast, **4g** and **4i** exhibited lower reactivity under the tested conditions and no apparent hydrolysis was observed in 6 days. These results suggest that structural modifications in the carbamate-D-ring affect the reactivity of the compounds. A comparison of the reactivity and germination-inducing potency of these compounds showed no direct correlation. (*R*)-**4a** and (*S*)-**4a** showed similar reactivity, but (*S*)-**4a** exhibited significantly higher germination-inducing

activity than (R)-**4a**; **4h** was far less reactive than **4g** and **4i**, while **4h** showed a slightly higher seed germination activity than **4i** (Supplementary Figs. 9 and 10).

Next, ShHTL7-mediated hydrolytic activity assays of **4a**, (R)-**4a**, (S)-**4a**, and **4g-i** were performed to understand the relationship with germination potency. The results showed that **4a**, (S)-**4a**, and rac-GR24 could be hydrolyzed by ShHTL7 by 36.4%, 40.1%, and 51.9%, respectively (Supplementary Fig. 11). In contrast, only 3.1% hydrolysis was observed for (R)-**4a**, showing that the configuration strongly affected the enzymatic hydrolysis. This was consistent with the hydrolysis of natural and synthetic SLs, in which, the (2′R)-isomers are more prone to hydrolysis by ShHTL7 than the (2′S)-isomers[26,40]. Structural modification of the 1′O in the D-ring of **4a** did not block the hydrolysis of **4g** and **4h** by ShHTL7, while changing the oxygen bridge to a methylene spacer between the carbonyl group and D-ring of the carbamate-D-ring of **4a** did block the hydrolysis of **4i** by ShHTL7

(Supplementary Fig. 11). Based on the above observations, we inferred that (S)-**4a** may have similar biochemical properties as 2′R-configured SLs. We then examined whether (S)-**4a** binding, just as for rac-GR24, would result in a covalently linked molecule in ShHTL7 by using Q-TOF mass spectrometry (SYNAPT-G2-Si, Waters company) analysis. The results showed that compared to the MS data of the recombinant ShHTL7 protein (Supplementary Fig. 12a), an increase by 94 Da was observed in (S)-**4a** treated ShHTL7 protein. For (R)-**4a** and **4g**, a weak peak (increase of 95 Da or 94 Da) of similar modification was found, respectively (Fig. 3). A similar modification was not observed in **4h** and **4i** treated samples (Supplementary Fig. 12b, c). To furtherly identify the specific modification of the increased mass of 94 Da or 95 Da, we incubated (R)-**4a**, (S)-**4a**, and **4g** with ShHTL7, followed by gel digestion and MS/MS analysis. We found that the $m/z = 94$ or 95 Da was modified on a ShHTL7 peptide with an amino acid sequence from 231 to 262. Further analysis showed that this mass

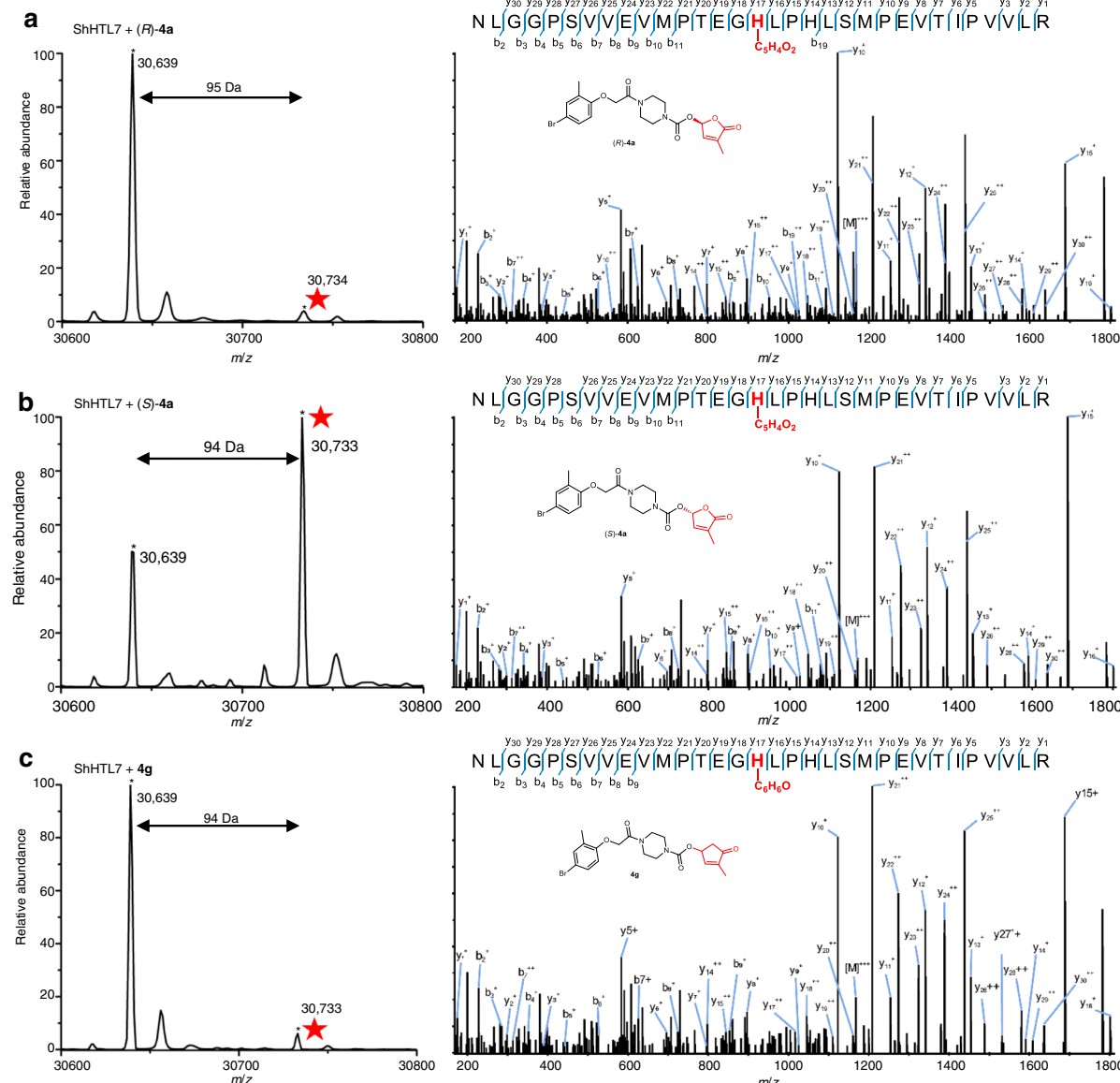

**Fig. 3 Analysis of the potential covalent modification on His246 of the catalytic triad. a–c** LC-MS analysis of the full-length protein (left) and LC-MS/MS analysis of the modified peptide (231-NLGGPSVVEVMPTEGHLPHLSMPEVTIPVVLR-262) (right) of ShHTL7 treated with (R)−**4a** (**a**), (S)-**4a** (**b**), and **4g** (**c**), respectively. The modified peptide at $m/z = 30,734$ or 30,733 on the left panels together with the MS2 spectrum on the right suggested that $C_5H_5O_2$ (**a**, **b**) or $C_6H_6O$ (**c**) was modified on the His246 of the catalytic triad, respectively. Labelled peaks on the right panels correspond to masses of y and b ions of the modified peptide.

corresponding to $C_5H_5O_2$ or $C_6H_6O$ was specifically covalently bound to His246 for (S)-**4a** and (R)-**4a**, or **4g**, respectively (Fig. 3a–c). No covalently linked molecule presence was identified in **4h** treated ShHTL7 (Supplementary Fig. 12b). We assume that this is due to the fact that the $N$-CH$_3$ D-lactam modified His246 is more reactive, resulting in quick decomposition of the intermediate. Collectively, our experiments demonstrate that the highly bioactive $S$-configuration (2′$S$) of **4a** is more prone to covalently linked molecule generation in ShHTL7, compared with the $R$-isomer. Additionally, our results also suggest that hydrolysis of the compounds by the receptor is not a determinant for signal transduction and germination-inducing activity, but is important for high activity.

**Enzymatic hydrolytic activity is favorable for signal transduction.** Considering the unclear perception mechanism in root parasitic weeds, we performed a yeast two-hybrid (Y2H) assay to investigate whether there is a relationship between the bioactivity of **4a**, (R)-**4a**, (S)-**4a**, and **4g-i** and their potency in promoting the interaction of ShHTL7 with the downstream proteins. The results showed that at the concentration of 1 µM, **4a** and (S)-**4a** could strongly induce the interaction of ShHTL7 with SMAX1, AtMAX2, and ShMAX2, similar to rac-GR24 treatment (Fig. 4a). In contrast, for (R)-**4a**, no ShHTL7-SMAX1 or ShHTL7-AtMAX2 interactions were detected at 1 and 5 µM. At 200 µM; however, (R)-**4a** did induce the interaction as rac-GR24 induced at 0.01 µM. Even at a concentration as low as 0.01 µM, **4a**, and (S)-**4a** could still promote the ShHTL7-ShMAX2 interaction. We found that **4h** exhibited a similar induction of protein interaction as (R)-**4a** at 1 and 200 µM, while for **4g**, only weak ligand-induced interactions between ShHTL7 and SMAX1 or AtMAX2 were observed at 200 µM (Supplementary Fig. 13). Additionally, almost no ligand-induced interactions were observed for **4i**, even at 200 µM. Together with our results from mass spectrometry

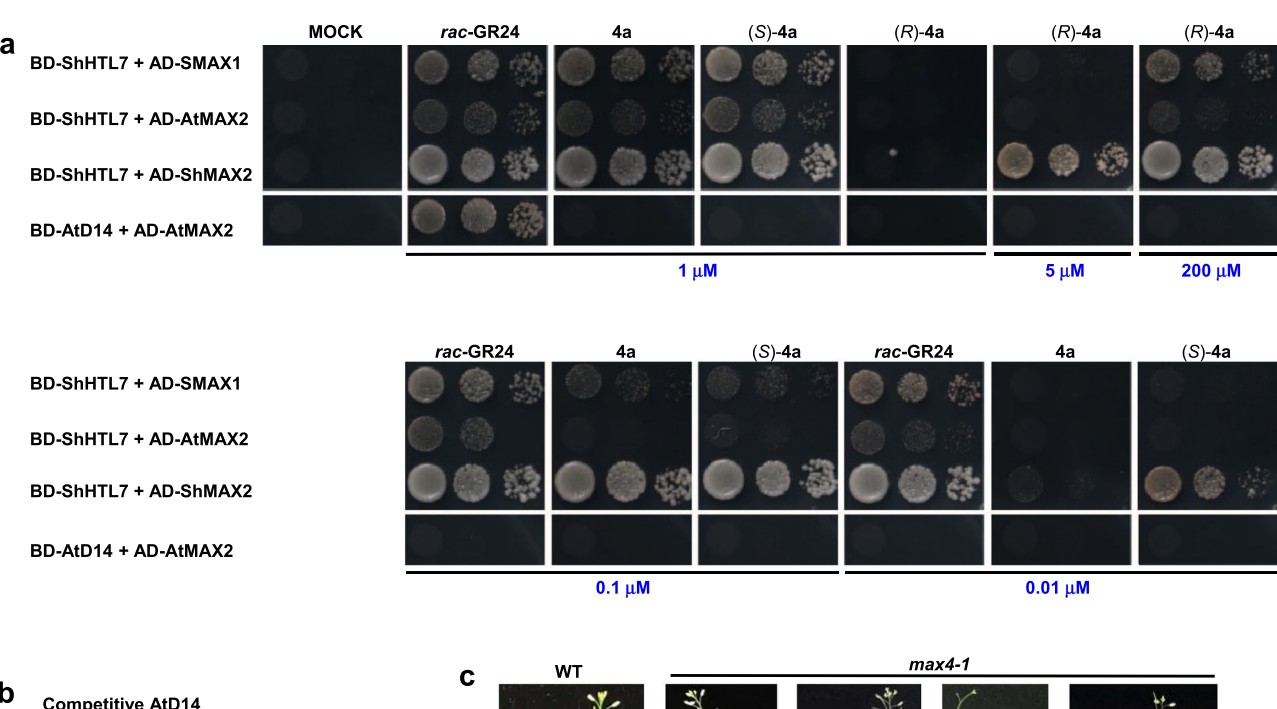

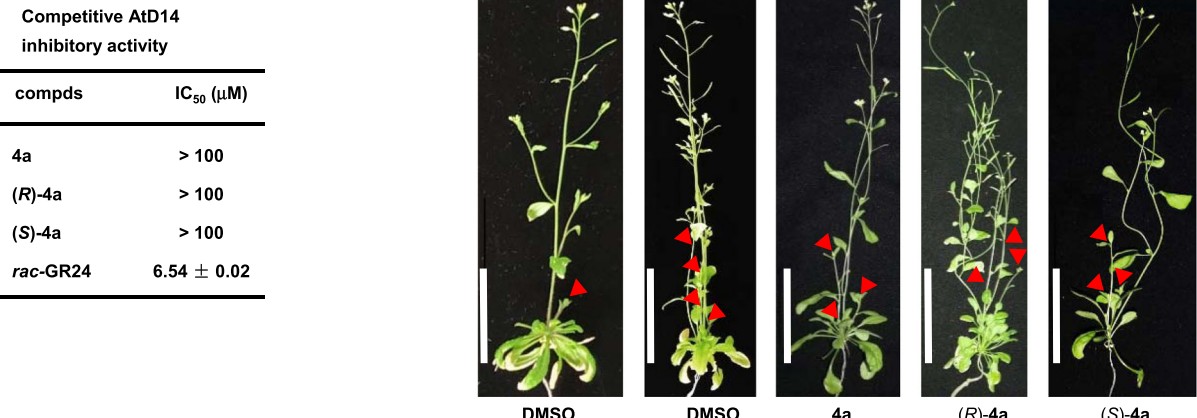

**Fig. 4 Bioassays with compounds 4a, (R)-4a, and (S)-4a. a** Yeast two-hybrid assays for ShHTL7 and AtD14 interactions with SMAX1, AtMAX2, ShMAX2, respectively. ShHTL7 and AtD14 were fused to GAL4-BD, SMAX1, AtMAX2, ShMAX2 were fused to GAL4-AD. Serial 10-fold dilutions of yeast cultures were spotted onto a selective growth medium that was supplemented with different concentrations of rac-GR24, **4a**, (R)-**4a**, and (S)-**4a**, respectively. Images show growth after 3 days on SD/-Leu-Trp-His-Ade plates at 30 °C. **b** Competitive AtD14 inhibitory activity of **4a**, (R)-**4a**, and (S)-**4a** using YLG as the substrate. Error bars mean SD ($n = 3$ biologically independent experiments). **c** **4a**, (R)-**4a**, and (S)-**4a** could not suppress the shoot branching phenotype of *max4-1*, an SL biosynthetic mutant, at 1 µM. The red arrowheads indicate axillary branches. No. of axillary shoots (over 0.5 cm) were shown as means ± SD ($n = 3$ biologically independent experiments). Scale bar, 5 cm.

analysis and bioassays (Figs. 2, 3), our study suggests that the high hydrolytic activity of the SL receptors for some agonists are favorable for SL signal transduction *in planta*, resulting in a high root parasitic weed seed germination-inducing activity.

Unlike *rac*-GR24, **4a** and its two enantiomers did not promote the interaction between AtD14 and AtMAX2, even at a concentration of up to 200 μM (Fig. 4a). We then examined the competitive inhibition $IC_{50}$ values of **4a** and its isomers on AtD14 by using YLG as the substrate. The results showed that the inhibitory activity of **4a**, (*R*)-**4a**, and (*S*)-**4a** towards AtD14 is relatively weak, with $IC_{50}$ values above 100 μM (Fig. 4b, Supplementary Fig. 14). To evaluate this result also in vivo, we investigated whether **4a** and its enantiomers affect the growth of *Arabidopsis*. We observed that at 1 μM, **4a**, (*R*)-**4a**, and (*S*)-**4a** failed to increase the root hair length in the wild-type (Supplementary Fig. 15) or suppress the enhanced shoot branching of *more axillary growth 4-1* (*max4-1*) (Fig. 4c). This result shows that the parasitic plant seed germination-inducing activity of these chemicals is selective and specific to ShHTL7 for the induction of downstream signal transduction.

**(*S*)-4a induces a higher ratio of privileged conformation of the ShHTL7-(*S*)-4a complex than (*R*)-4a.** To interpret the high stereospecificity of ShHTL7 in hydrolyzing (*R*)-**4a** and (*S*)-**4a**, we invested substantial efforts, using co-crystallization, to capture (*R*)-**4a** and (*S*)-**4a** inside ShHTL7. However, none of the obtained crystal structures contained an intact (*R*)-**4a** or (*S*)-**4a**, or even a hydrolytic fragment of the two molecules. We hypothesize that the reason for this is that ShHTL7 can simultaneously bind and cleave (*R*)-**4a** and (*S*)-**4a**, which prevents the formation of the receptor-ligand complex during crystallization. We, therefore, performed molecular simulation studies of (*R*)-**4a** and (*S*)-**4a** with the obtained apo ShHTL7 (PDB code: 6A9D, Supplementary Fig. 16, Supplementary Table 2). Our computational investigation indicated that there were no significant differences between the binding modes of (*R*)-**4a** (Fig. 5a) and (*S*)-**4a** (Fig. 5b) at the atomic level. Both (*R*)-**4a** and (*S*)-**4a** could form favorable hydrogen-bonding interactions with Y26 and M96, respectively. In contrast to (*S*)-**4a**, there were two possible binding modes of (*R*)-**4a** (Fig. 5a) with ShHTL7. Hence, this similar static in silico docking structures cannot explain the influence of the absolute configuration of **4a** on the level of hydrolysis.

Next, we compared the electrostatic potential (ESP) surfaces and Mulliken atomic charges of (*S*)-**4a** by using Density Functional Theory (DFT) calculation. We observed that the negative ESP surfaces were mainly located around the oxygen atoms of the carbamate-D-ring functional groups of (*S*)-**4a** (Supplementary Fig. 17). The Mulliken charge of 2′, 2″, 3″, and 5′ carbon atoms of (*S*)-**4a** were 0.312, 0.583, 0.831, and 0.579 a.u., respectively (Supplementary Fig. 18), suggesting that these atoms are more inclined to be attacked by the nucleophilic amino acids during the enzymatic reaction. It has been proposed that the hydrolysis of SLs is initiated by the nucleophilic attack of the Serine residue to the carbonyl group of the D-ring of the catalytic triad (Supplementary Fig. 19a)[13,23,26,27]. Indeed, mutations showed that the ShHTL7[S95A] and ShHTL7[H246A] mutants were devoid of hydrolase activities compared with the wild-type enzyme (Supplementary Fig. 20a), and Seto et al. have reported that AtD14[D218A] also lost its hydrolytic activity[40]. These data demonstrate that the integrity of the catalytic triad is indispensable for hydrolysis by the receptor.

To clarify the dynamics of how the catalytic triad affects hydrolysis by ShHTL7, we performed MD simulations to study the differences between (*R*)-**4a** and (*S*)-**4a** (Supplementary Fig. 21). After 1000 ns MD simulations, we found that both Conf1 and

Conf2 of (*R*)-**4a** had undergone a conformation transition (Fig. 5c, d), while the conformation change of Conf2 was larger than that of Conf1. The carbamate-D-ring warhead of Conf2 has shifted away from the catalytic center, suggesting that Conf2 is a less favored conformation in the active site of ShHTL7. The binding conformation of (*S*)-**4a** did not change during the MD simulation (Fig. 5e), which might attribute to the strong hydrogen-bonding interaction between the M96 and the D-ring, which in turn stabilized the binding conformation. To further characterize our computational models, we performed mutation studies of the key residues in the active pockets that may have interactions with our molecules. Indeed, a number of the mutations of specific residues altered the $IC_{50}$ values, supporting our simulated models (Supplementary Fig. 20b).

According to the collision theory, an appropriate distance exists between a ligand and the catalytic site of the enzyme that correlates with the enzymatic hydrolysis. The S95-H246-D217 triad represents the most important catalytic residues of ShHTL7. The hydrogen-bond interactions of the three residues not only stabilize the conformation of the catalytic triad but also increase the nucleophilicity of the OG atom of S95[23]. Therefore, we used the distances of 5′C and OG atoms ($D_{5'C-OG}$), NE2 and HG atoms ($D_{NE2-HG}$), OD1 and HD1 atoms ($D_{OD1-HD1}$), and OD2 and HD1 atoms ($D_{OD2-HD1}$) as a set of geometry parameters to describe the features of the conformational distribution of the ShHTL7-mediated hydrolysis process of (*R*)-**4a** and (*S*)-**4a** (Fig. 5f). After analyzing the distances of $D_{OD1-HD1}$ and $D_{OD2-HD1}$ from the MD trajectories, we found that the hydrogen-bond length for H246 and D217 of the ShHTL7-(*R*)-**4a** (Conf1) (Supplementary Fig. 22a), ShHTL7-(*R*)-**4a** (Conf2) (Supplementary Fig. 22b), and ShHTL7-(*S*)-**4a** (Supplementary Fig. 22c) systems were all around 2 Å, indicating that the hydrogen-bond interactions of H246 and D217 did not influence the protein hydrolysis of (*R*)-**4a** and (*S*)-**4a**. A lot of research has shown that the geometry distances for the reaction between substrate and enzyme generally varied from 1.5 to 3.8 Å[41,42]. After carefully examining the X-ray crystal structures of the α/β-fold hydrolase SL receptors, we found that the distances of HG of Ser and NE2 of His were generally within 1.6–3.5 Å (Supplementary Table 3), suggesting that these hydrogen bonds were formed during the crystallization[43]. Based on the above discussion, we defined the conformation with distances of $D_{5'C-OG}$ within 1.5–3.8 Å and $D_{NE2-HG}$ within 1.6–3.5 Å as the privileged conformation (PC). The appropriate collision probability between ShHTL7 and a ligand was quantified as $P_{PC}$. By using our previous Prenzyme method[42], the $P_{PC}$ of the ShHTL7-(*R*)-**4a** (Conf1), ShHTL7-(*R*)-**4a** (Conf2), and ShHTL7-(*S*)-**4a** complexes were calculated as 1.3%, 0%, and 33.0% respectively (Fig. 5g), which correlates well with the levels of ShHTL7-mediated hydrolysis (Supplementary Fig. 11). Collectively, our results indicate that the absolute configuration of **4a** affects the hydrolytic activity of ShHTL7 by inducing ShHTL7 to preferentially acquire the privileged conformation.

Additionally, according to our DFT calculations, we found that the Mulliken charge of the 2′C atom of (*S*)-**4a** was 0.312 a.u., suggesting that this atom can also be attacked by the nucleophilic residues during the enzymatic reaction. Therefore, we performed another privileged conformation statistics analysis and defined the distance between the 2′C atom of (*S*)-**4a** and NE2 atom of H246 as $D_{2'C-NE2}$ of 1.5–3.8 Å. $D_{2'C-NE2}$ and $D_{NE2-HG}$ were then used as geometry parameters for describing the privileged conformation. The $P_{PC}$ was calculated as 19.2%, suggesting that this may represent another possible pathway of ShHTL7-mediated (*S*)-**4a** hydrolysis. In this hydrolytic pathway, the hydrogen-bonding interactions of Thr157 and 2O atom, and Tyr174 and 3O atom could activate the leaving group (Supplementary Table 4)[44], the NE2 atom of His246 directly

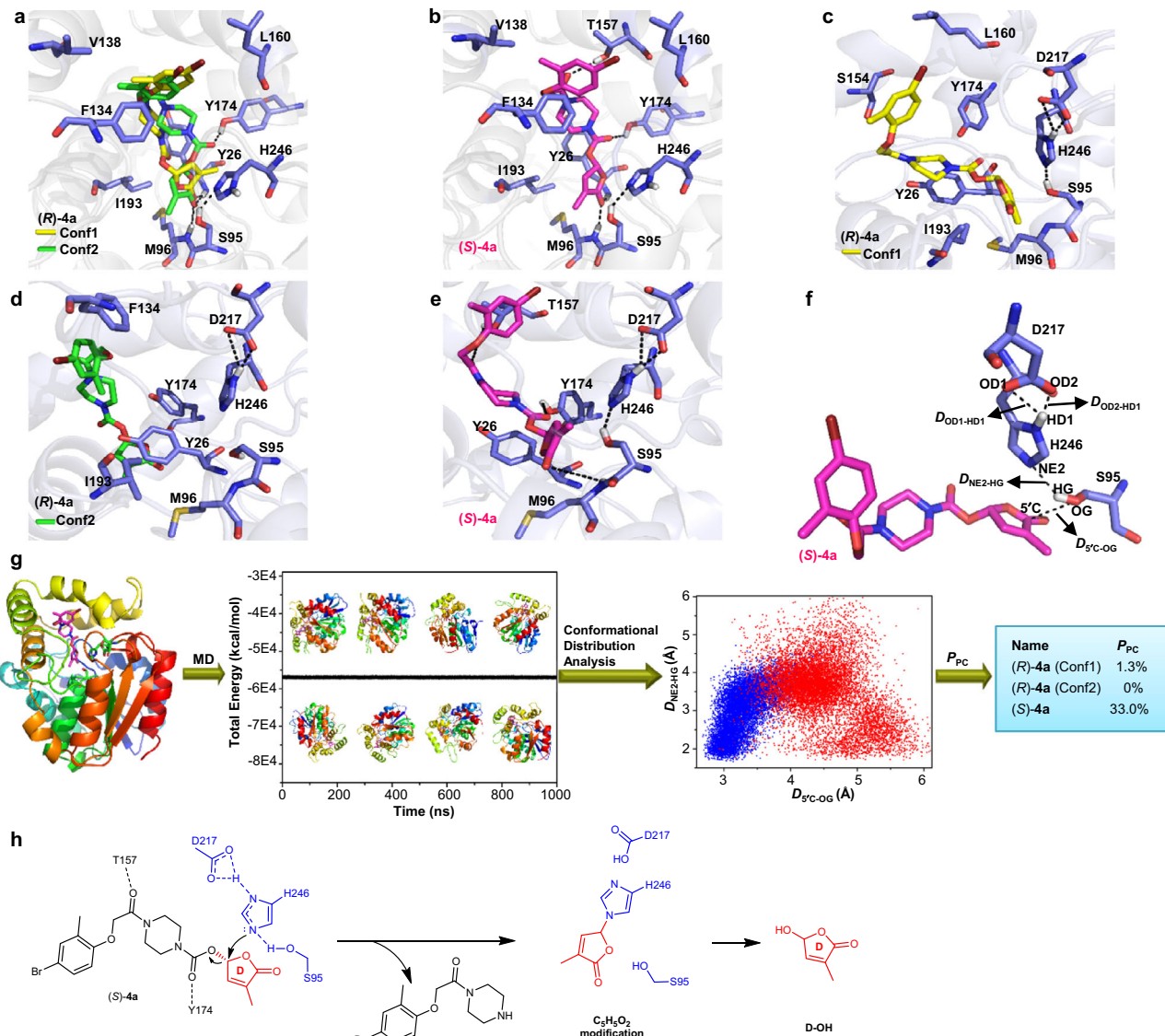

**Fig. 5 Molecular simulations of compounds (R)- and (S)-4a with ShHTL7.** The key residues around the binding pocket are shown as blue sticks and the hydrogen-bonding interactions are shown as black dashed lines. **a** Docking models of (R)-**4a** to ShHTL7, there were two reasonable binding modes: Conf1 (shown in yellow stick) and Conf2 (shown as green stick). **b** Docking model of (S)-**4a** to ShHTL7, (S)-**4a** is shown as magentas sticks. **c–e** Binding models of (R)-**4a** (conf1), (R)-**4a** (conf2), and (S)-**4a** to ShHTL7, respectively. The models were taken from the last frame of each simulation at 1000 ns. **f** A schematic description of the hydrolysis reaction of the catalytic triad. $D_{5'C-OG}$, $D_{NE2-HG}$, $D_{OD1-HD1}$, and $D_{OD2-HD1}$ were used as a set of geometry parameters to describe the privileged conformation (PC) of ligands and ShHTL7. **g** A process to predict the privileged conformation of (R)- and (S)-**4a** with ShHL7 based on the MD simulations and conformational distribution statistical analysis. **h** A proposed alternative schematic diagram of the ShHTL7-mediated (S)-**4a** hydrolysis pathway. The hydrogen-bonding interactions of Thr157 and 2O atom, and Tyr174 and 3O atom could activate the leaving group, and the catalytic process begins with the nucleophilic attack by the NE2 atom of His246 to the 2'C atom of (S)-**4a**, which results in the $C_5H_5O_2$ modification of His246.

attacks the 2'C of the D-ring of (S)-**4a** and makes the $C_5H_5O_2$ modification on His246 (Fig. 5h, Supplementary Fig. 23). Furthermore, our MS-MS analysis of the $C_6H_6O$ modification on ShHTL7 by **4g** also supports that the hydrolysis can proceed via nucleophilic attack of the NE2 atom of H246 to the 2'C atom of the D-ring, resulting in the $C_6H_6O$ modification on H246 (Fig. 3c, Supplementary Fig. 19b, c). In addition, we have demonstrated that the nucleophilic attack of the imidazole group of N-acetyl-L-histidine methyl ester to 2'C atom of **4g** under mild reaction conditions is also possible (Supplementary Table 5). We also observed that **4g** could be hydrolyzed by ShHTL7$^{S95A}$ and ShHTL7$^{S95C}$ mutant proteins (Supplementary Fig. 24a). The LC-MS analysis showed that a molecule with m/z around 95, which

corresponds to the modified D-ring of **4g**, is covalently linked to the ShHTL7$^{S95A}$ and ShHTL7$^{S95C}$ proteins (Supplementary Fig. 24b–f). Thus, we revealed an unexpected mechanism of ShHTL7-mediated SL hydrolysis.

## Discussion

The development of suicidal germination inducers to control parasitic weeds is hampered by insufficient insight into the SL perception mechanism. Here, a SL receptor agonist scaffold was identified by using a virtual-screening strategy based on the most sensitive SL receptor ShHTL7. The lead compound from this screen was systematically modified through structure-guided

design and scaffold hopping approaches to create (S)-**4a**, a highly bioactive *P. aegyptiaca* and *S. hermonthica* seed germination stimulator, with $EC_{50}$ value as low as $3.39 \times 10^{-11}$ M, and $1.93 \times 10^{-17}$, respectively. Although (S)-**4a** was hydrolyzed by ShHTL7 similarly as *rac*-GR24, and also resulted in $C_5H_5O_2$ modification of His246 of (S)-**4a** treated ShHTL7 protein, (S)-**4a** displayed a lower reactivity than *rac*-GR24, and promoted ShHTL7-ShMAX2 interaction in a concentration as low as 10 nM. Finally, we found that (S)-**4a** induced ShHTL7 to preferentially acquire a more privileged conformation than (R)-**4a**, which agrees with the *R*-configured D-ring occurring in the natural SLs.

Based on our experimental evidence, we propose three but coexisting perception mechanisms by the SL receptors for synthetic SL agonists (Fig. 6). The first route is represented by ShHTL7 binding hydrolysis-resistant agonists, such as **4i**, in its open conformation, which activates downstream SL signal transduction. However, signal transduction through this pathway was very low in efficacy, as evidenced by weak interaction between ShHTL7 and ShMAX2 in the Y2H assay and the low seed germination stimulant activity. The second route is represented by agonists that can be hydrolyzed by ShHTL7; our results suggest that molecules, such as (S)-**4a**, resulting in a favorable CLIM modification of ShHTL7; they show significantly improved potency in triggering SL signal transduction than compounds, such as (R)-**4a**, and **4g**, that are less prone to result in CLIM modification. Although the above perception mechanisms can be verified by our seed germination assays, ShHTL7-hydrolysis experiments, and Y2H assays, it is unclear whether the conformation of ShHTL7 was changed or not, after the compound was hydrolyzed but did not result in CLIM modification on the catalytic triad. We, therefore, conclude that the efficiency of synthetic SL agonists to activate SL signaling is largely depends on their hydrolyzable and covalently linked molecule modification.

Our results show that ShHTL7 has a high ligand stereoselectivity. First, (S)-**4a** showed 36-fold stronger competitive ShHTL7 inhibition activity than (R)-**4a** (Fig. 2b); second, the hydrolysis ratio of (S)-**4a** by ShHTL7 was much higher than that of (R)-**4a** (Supplementary Fig. 11); third, (S)-**4a** was more efficient than (R)-**4a** in promoting the interaction of ShHTL7 with SMAX1, AtMAX2, and ShMAX2 (Fig. 4a); last, by combining MD simulations and statistical analysis, we found that (S)-**4a** can induce ShHTL7 to have a higher ratio of privileged conformation than (R)-**4a** (Fig. 5g). Our observations can also explain the high bioactivity of 2′*R*-configured synthetic SL analogs[17]. de Saint Germain et al. reported that 2′*R* GC242 showed higher activity than 2′*S* GC242[26]. We found that (S)-**4a** exhibited higher bioactivity than (R)-**4a**. Structural superimposition of 2′*R* GC242 and (S)-**4a** to GR24[5DS] show that there is a high similarity between the core structures (O-D-ring) in these three compounds (Supplementary Fig. 25). Thus, stereoisomers with an O-D-ring moiety that well overlays with that of natural SLs would show higher potency. Additionally, another possible pathway for ShHTL7-mediated ligand hydrolysis was proposed based on the MD simulations, MS-MS analysis, and S95 mutation studies. That is, the reaction may be initiated through the nucleophilic attack of the 2′C atom of the D-ring by the NE2 atom of His246 (Fig. 5h, Supplementary Fig. 19b, c).

Although the series of aryloxyacetyl piperazines were initially obtained from an ShHTL7-based virtual screening, these molecules exhibited improved potency in stimulating the germination of both *S. hermonthica* and *P. aegyptiaca* seeds. Additionally, agonists with a carbamate-D-ring scaffold, showing higher competitive ShHTL7 inhibition activity, also exhibited improved seed germination-inducing activity in both *S. hermonthica* and *P. aegyptiaca*. In *P. aegyptiaca*, 5 KAI2s have been proposed as the SL receptors for the regulation of seed germination[12]. From sequence alignments, we found PaKAI2s showed more than 52% and 62% sequence

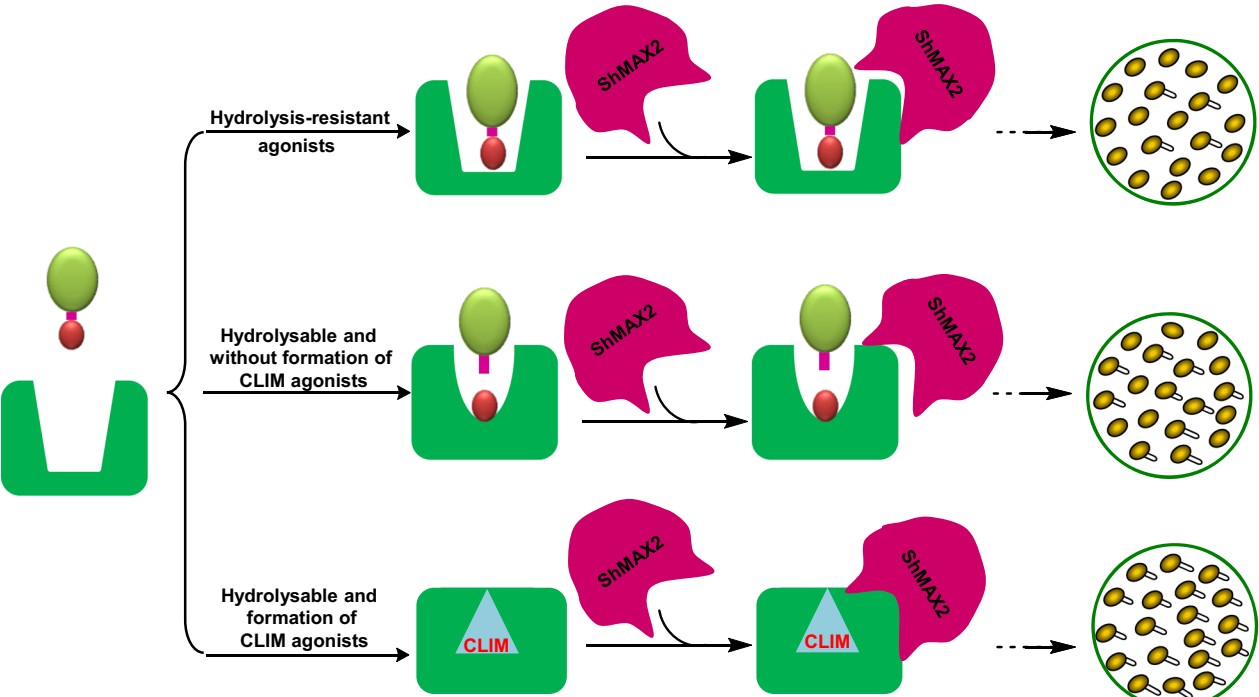

**Fig. 6 A proposed structure–perception relationship model of synthetic SL agonists by ShHTL7 in the signal transduction pathway.** For hydrolysis-resistant agonists (i.e., compound **4i**), ShHTL7 might bind with the intact molecule in the open conformation and trigger downstream signaling; however, the efficiency is rather low. For hydrolyzable agonists, compounds that are less prone to result in CLIM modification of ShHTL7 showed lower potency in activating the signal transduction than compounds that are more prone to result in CLIM modification.

similarity with ShHTLs and ShHTL7, respectively (Supplementary Fig. 26). The high sequence similarity may explain the consistency of these compounds in inducing germination in both *S. hermonthica* and *P. aegyptiaca*. Additionally, the YLG-based competition assay for the 10 HTL proteins in *S. hermonthica* and 5 KAI2s protein homologs in *P. aegyptiaca* also suggest that **4a** and (*S*)-**4a** bind to these proteins, with $IC_{50}$ values ranging from 0.94 to 55.76 μM (Supplementary Fig. 27). Collectively, we identified highly effective seed germination stimulants that can potentially be used for the control of root parasitic weeds, and as probes to further study the SL perception mechanisms in these root parasitic plant species.

## Methods

**Virtual screening**. The structures of 13 representative molecules were constructed by Sybyl 6.9 (Tripos, Inc.)[45]. An automatic tool in DS 2.5 (Accelrys Inc) was used for the hierarchical 2D molecular fingerprint similarity-based virtual screening[35]. A Specs database contained 212,739 commercial compounds that were used for the screening. For each molecule, 100 most similar compounds were screened and clustered by DS 2.5, and the cluster center with 10 compounds was then selected for further analysis. The candidates were selected based on the chemical diversity and physicochemical property analysis, and purchased from Specs for enzymatic assay.

**Protein overexpression and purification**. ShHTLs, AtD14, and PaKAI2s genes were synthesized and cloned into the pET-28b plasmid, respectively, by WuXi Qinglan Biotech co., Ltd (YiXing, China). Then, Plasmid DNA was extracted following the protocol of Endofree Plasmid Extraction Mini Kit (CWBIO, Beijing, China). The construction of mutants used in this study was performed following the site-directed mutagenesis method based on ShHTL7 plasmid DNA[46]. For protein overexpression, the exacted plasmids were transformed into *E. coli* BL21 strain, respectively. The obtained single clone was used to perform protein overexpression and purification. When cells were grown to an $OD_{600}$ of 0.6–0.7, 0.5 mM isopropyl *β*-Dthiogalactopyranoside (IPTG) was added to induce the protein overexpression. After inducing at 18 °C for 12 h, cells were harvested, lysed, and purified by Ni-NTA column (QIAGEN) with elusion buffer (50 mM HEPES pH 7.5, 150 mM KCl, 10 mM $MgCl_2$, 100 μM DTT, 250 mM imidazole, 10% glycerol (v/v), and 10 mM EDTA)[23].

**Surface plasmon resonance (SPR) analysis**. The interactions of small molecules with recombinant ShHTL7 were evaluated using the Biacore T200 system (GE Healthcare, USA) at 25 °C[47]. The purified ShHTL7 protein was immobilized on a CM5 sensor chip using the standard amine coupling procedure. The immobilization level of ShHTL7 was about 14,500 resonance units (RU). PBS-P (10 mM phosphate buffer, 2.7 mM KCl, 137 mM NaCl, 0.05% Surfactant P20) containing 0.5% DMSO was used as the running buffer. For the screening experiments, candidates with different structures at 50 μM in the running buffer were injected individually at a flow rate of 30 μL min$^{-1}$. For binding affinity studies, compounds with different concentrations in running buffer were used. During the experiments, both the contact and dissociation times were set as 1 min. Data analysis was performed using the Biacore Evaluation Software (T200 Version 2.0). The affinity values ($K_D$) were obtained by the kinetics analysis or the steady-state affinity method.

**Yeast two-hybrid assays**. To construct plasmids for yeast two-hybrid assays, the CDS of ShHTL7 and AtD14 were cloned into yeast expression vector pBrigde to generate BD-ShHTL7 and BD-AtD14, and the CDS of ShMAX2, AtMAX2, SMAX1 were cloned into pGADT7 to make Gal4 DNA activation domain (AD) constructs, respectively[23]. Primers used for yeast expression vector construction are listed in Supplementary Table 6. Yeast two-hybrid assays were performed using the Yeastmaker Yeast Transformation System 2 (Clontech). In brief, Yeast AH109 cells were co-transformed with specific bait and prey constructs and coated on selective growth medium SD/-Leu-Trp for 3 days at 30 °C, pick the positive constructs into liquid selective growth medium SD/-Leu-Trp shaking for 36 h at 30 °C. Washed yeast cells three times and diluted, made sure OD600 reached 2.5, then serial 10-fold dilutions of yeast cultures were spotted onto selective growth mediums that were supplemented with different concentrations of *rac*-GR24 and tested molecules, respectively. All yeast transformants were grown on a selective growth medium at 30 °C for 3 days.

**Hydrolysis assays with Yoshimulactone green (YLG)**. According to the reported methods[22,28], the competitive inhibition of SL receptors (ShHTLs, AtD14, and PaKAI2s) activity by small molecules was performed by using YLG as the substrate at 25 °C. The excitation and emission wavelengths of YLG are 480 and 520 nm, respectively. We used a fluorescence spectrophotometer (CARY Eclipse) to detect the fluorescence intensity change of YLG during the enzymatic hydrolysis

assay process. All the testing compounds and YLG were dissolved in DMSO and prepared as stock solutions just before use. A total of 100 μL reaction mixture, which consisted of 4–6 μg recombinant proteins, 3 μM YLG in reaction buffer (100 mM HEPES, 150 mM NaCl, pH 7.0), and 0.0256–400 μM compounds were added to a 96-well black plate (Thermo). The changes of fluorescence at 520 nm of each well were recorded every 1 min for 30 min. Using the reaction mixture without adding enzyme and tested compounds as the background control. After subtracting the background hydrolysis, a linear fluorescence increase in 8 min was used for further inspection. The $IC_{50}$ values were determined via fitting relative inhibition percentages with the varying concentrations of compounds in Microsoft Excel.

**S. hermonthica seed germination assays**. Two hundred mg of *Striga hermonthica* seeds were surface sterilized in 2% sodium hypochlorite containing 0.02% Tween-20 for 5 min, and then washed five times with sterile MilliQ water. The sterilized *Striga* seeds were spread on sterile glass fiber-filter papers (Whatman® GF/A, Sigma–Aldrich) in Petri dishes containing filter paper moistened with 3 mL sterile MilliQ water. The Petri dishes were incubated 6–8 days at 30 °C for pre-conditioning after which the seeds were dried in a laminar sterile air flow for 2 h. The germination bioassay was performed in 0.1% agarose and 0.1 ppm *rac*-GR24 was used as a positive control. Stocks of *rac*-GR24 and other compounds were prepared in acetone and diluted before use. Dried preconditioned seeds were added to the agarose containing the relevant compound, which was subsequently divided over three wells of a 12-well Cell Culture Treated Plates (Fisher Scientific, Landsmeer, the Netherlands). The plates were sealed and incubated in the dark at 30 °C for 2 days. The *Striga* germination rate in % (GR%) was calculated for each replicate using the formula: GR% = (Ngs/Nts) × 100, where Ngs is the number of germinated seeds per well and Nts is the total number of seeds per well. $EC_{50}$ values of the compounds were calculated using http://www.ic50.tk/index.html.

**P. aegyptiaca seed germination assays**. *P. aegyptiaca* seeds were harvested from Xinjiang province, China in 2017. The seeds were surface sterilized using 1% NaClO solution for 2 min and then washed with distilled water six times. For seed germination assay, about 20 pieces of glass fiber-filter paper discs (15 mm diameter, Whatman GF/B) were placed separately on two layers of filter papers (9 cm diameter, Whatman) in a Petri dish. Five milliliters of distilled water was added to each Petri dish to wet the papers. Approximately 40–90 sterilized seeds were spread onto each glass fiber-filter paper disc. After the seeds were conditioned in the dark at 25 °C for 4 days[48], glass fiber-filter paper discs were blotted dry on filter papers (9 cm) and transferred to the wells of 24-well cell culture plates. The testing compounds were dissolved in acetone and diluted with $H_2O$ ($V_{H_2O}/V_{acetone}$ = 999/1) just before use[49]. *rac*-GR24 and 1‰ acetone solution were used as positive and negative controls, respectively. One hundred microliters of the test solution was added to each well of the plates, and all treatments were performed in three biological repeats. The plates were sealed and incubated in the dark at 25 °C for a week. The experiments were performed at least three times. The germinated seeds were observed by using an electronic magnifier at ×20 magnification. $EC_{50}$ values of the highly potent compounds were calculated using http://www.ic50.tk/index.html.

**Chemical reactivity analysis**. Compounds **4a**, (*R*)- and (*S*)-**4a**, **4g-i**, and *rac*-GR24 were dissolved in acetone as a stock solution (5 mmol) just before the test[50,51]. One hundred twenty microliters of sample solution was added to a solution of PBS (10 mM, pH 7.4, 810 μL) and methanol (270 μL). The solution was stirred vigorously for 30 s and then transferred to HPLC vial, and the vial was stored at 26 ± 1 °C. The remaining compounds in the vials were analyzed by using Agilent Technologies 1260 HPLC system every 1 h. The analytical column was Extend-C18 (4.6 mm × 150 mm × 5 μm). The mobile phases were $CH_3CN$ and $H_2O$ (containing 0.1% $HCO_2H$), $V_{CH3CN}:V_{H2O}$ = 50:50, with a flow rate of 1 mL/min, and an injection volume of 5 μL. The percentage of remaining compounds in the solution was calculated from the peak area. The half-life times ($t_{1/2}$) of samples were calculated.

**ShHTL7-hydrolysis reaction analysis**. Compounds **4a**, (*R*)- and (*S*)-**4a**, **4g-i**, and *rac*-GR24 were dissolved in DMSO and prepared as stock solution (1 mmol) just before use. A 200 μL of the reaction solution consisting of PBS buffer (pH 7.3), 10 μM samples, and 10 μg recombinant ShHTL7 or its mutants was incubated at 30 °C for 30 min; with the reaction solution (200 μL) containing PBS buffer (pH 7.3) and 10 μM samples as the blank control[40,52]. Subsequently, the reaction was terminated by adding 200 μL of methanol, and then the reaction mixture was shaken for 10 s and centrifuged at 5310 × *g* for 60 s. The supernatant was filtered and the remaining amount of samples was analyzed by HPLC. The relative percentages of hydrolysis were obtained by using the amount of compounds in blank control as reference.

**Mass spectrometric analysis of ligands and ShHTL7 or its mutants reaction systems**. For LC-MS analysis: a 200 μL reaction solution (100 mM HEPES, 150 mM NaCl, pH 7.0) containing 10, or 400 μM of tested samples **4a**, (*R*)- and (*S*)-**4a**, and **4g-i**, and 100 μg of recombinant ShHTL7 or its mutants was incubated at 25 °C for 30 min. Subsequently, the solution was filtered and analyzed by a

nanoACQUITY UPLC system, which was directly interfaced with an SYNAPT-G2-Si mass spectrometer produced by Waters Company. For LC-MS/MS analysis, the above chemicals treated ShHTL7 reaction solution was subjected to the SDS-PAGE analysis. The corresponding protein band around 30 KDa was excised, respectively, then digested with trypsin (Promega) in $NH_4HCO_3$ solution (50 mM) at 37 °C for 24 h[23,26–28]. The covalent modification of $C_5H_5O_2/C_6H_6O$ on the peptide was analyzed by a Thermo-Dionex Ultimate 3000 HPLC system, which was directly interfaced with a Thermo Orbitrap Fusion Lumos mass spectrometer. MS-MS spectra are generated by pLabel software[53,54].

***Arabidopsis* phenotype assays**. For *Arabidopsis* root hair assay, *Arabidopsis thaliana* (Col-0) seeds were surface sterilized by using 1% NaClO solution for 60 s and 75% ethanol solution for 60 s, and then washed by sterilized distilled water five times. The sterilized seeds were sowed on 1/2 MS media (2.202 g/L Murashige and Skoog basal salts, 0.5 g/L MES, 10 g/L sucrose, and 10 g/L agar, pH 5.7) which contained indicated concentrations of compounds. 0.1% DMSO was used as a negative control. The plates were placed 4 days at 4 °C and then placed vertically under fluorescent white light (100–150 µmol m$^{-2}$ s$^{-1}$) with 16 h light and 8 h dark at 20 ± 1 °C for 15 days[17,55]. The root hairs were imaged by OLYMPUS (U-HGLGPS), and the lengths were analyzed by ImageJ. The experiments were repeated three times. For *Arabidopsis* shoot branching assay: Col-0 and *max4-1* seeds were used in this experiment. Seeds surface sterilization was sterilized as described abouve. The *Arabidopsis* seedlings were placed on top of the cut black microcentrifuge tubes filled with 1/2 MS. The tubes were placed 4 days at 4 °C and then moved to fluorescent white light (100–150 µmol m$^{-2}$ s$^{-1}$) with 16 h light and 8 h dark at 20 ± 1 °C for 7 days[26]. The germinated seedlings were grown on hydroponic culture solutions (containing indicated concentrations of **4a**, (*R*)- and (*S*)-**4a**), with the solution containing 0.01% DMSO as the negative control. The solutions were renewed every week. After 4 weeks, the numbers of rosette axillaries were analyzed.

**Crystallization**. The sitting-drop vapor diffusion method was used to obtain the crystal structure of ShHTL7. The details are shown in supporting information[52].

**Molecular simulation studies**. The 3D structures of (*R*)- and (*S*)-**4a** were constructed by Sybyl 6.9 (Tripos, Inc.). Autodock 4.2 was used for docking ligands into the structure of ShHTL7 (PDB entry 6A9D)[56]. MD simulations of ligand-receptor complexes were performed by using Amber 14. The results were analyzed by cpptraj[57]. The $P_{PC}$ calculation and MD details are shown in the supporting information. The structure figures were analyzed by PYMOL software[58].

**Reporting summary**. Further information on research design is available in the Nature Research Reporting Summary linked to this article.

## Data availability

The structure factors and atomic coordinates of ShHTL7 are available at the Protein Data Bank with the accession number: 6A9D. The crystallographic data for compounds (2*R*)-**3f**, (2*S*)-**3f**, and (2*S*)-**3g** are available at the Cambridge Crystallographic Data Centre (CCDC) with the accession number: 1856748, 1856749, and 1883470, respectively. Copies of the data can be obtained free of charge via https://www.ccdc.cam.ac.uk/structures/. Methods and all relevant data are available in Supplementary Information and from the authors. Source data are provided with this paper.

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

## Acknowledgements

We acknowledge the funding support by the National Key Research and Development Program of China (No. 2021YFD1700102, 2017YFD0200501, and 2021YFA1300400), the National Natural Science Foundation of China (No. 22077072, 21837001, 32070321, and 21740002), and China Hunan Provincial Science and Technology Department (No. 2019RS2019 and 2020JJ3007). We thank Prof. Yongqing Ma at Northwest A&F University for helping in seed germination test of *Phelipanche aegyptiaca*. We thank the National Supercomputer Center in Tianjin, and the calculations were performed on TianHe-1(A).

## Author contributions

Z.X. conceived and designed the project. D.W. performed the virtual screening and organic synthesis. Z.P. expressed and purified the proteins. Z.P. and J.W. performed SPR assay. Z.P. and S.Y. conducted enzymatic and hydrolytic assay, D.W., Z.P., H.Y., and R.Y. conducted mass spectrometry, H.Y. and R.Y. conducted Y2H analysis. B.T. and A.W. performed *S. hermonthica* seed germination assay. D.W. and X.W. performed MD studies. Z.P., S.Y., T.W., and L.L. conducted the plant growth and *P. aegyptiaca* seed germination assays. Y.Z. performed protein crystallization experiments. D.W., Z.P., H.J.B., R.Y., and Z.X. wrote the manuscript with inputs from all authors.

## Competing interests

The authors declare no competing interests.
