## [Peer Review File · Nature Communications]

Probing strigolactone perception mechanisms with rationally designed small-molecule agonists stimulating germination of root parasitic weedsREVIEWER COMMENTS

Reviewer #1 (Remarks to the Author):

This manuscript by Wang et al. presents a set of rationally designed agonists for ShHTL7 that are able to act as germination stimulants for broomrape and witchweed. Using these compounds, the authors also suggest that full signaling activity of the receptors depends on substrate hydrolysis. Additionally, given the difference in bioactivity toward ShHTL7 vs. AtD14, the reported 4a compound appears to be a promising candidate for selective agonism for parasitic weeds. The authors also present evidence that strigolactone hydrolysis may occur via an SN₂-like mechanism involving nucleophilic attack of H247 on the butenolide ring. While this is a very comprehensive and well-written manuscript overall, I have several suggestions for improvements. In particular, while the presented results (particularly on compound 4g) do indicate that the newly proposed reaction mechanism is possible, it is not convincing that this reaction would occur substantially in strigolactones or strigolactone analogs containing an intact butenolide ring.

Major Points

1. On pg. 6, line 126, the authors point out that "compounds with electron-withdrawing groups on the benzoyl moiety had a smaller KD than those with electron-donating groups." Since energetic contributions of substrate hydrolysis are often not distinguishable from energetic contributions of substrate binding in in vitro KD measurements, could this observation be explained by change in reactivity of the substrate?
2. On pg. 11, line 227-229, the authors state that the CLIM molecule was detected by Q-TOF mass spectrometry. The CLIM molecule is an open form of the butenolide ring that is covalently linked to H246 and S95 and thus would have the same mass as a butenolide ring. Is there additional data that attributes this to a CLIM molecule rather than a butenolide ring attached to H246 or S95?
3. On pg. 11, line 239-240, the authors state that they assume that the absence of CLIM is "due to the fact that the N-lactam modified His246 is unstable." This is a reasonable explanation, however, another contributing factor could be decreased electrophilicity of the adjoining 2'C and 5'C carbons vs. when oxygen is present in the same position on the substrate. Could the authors comment on this possibility?
4. While hydrolytic activity on compound 4g does suggest that the newly reported mechanism is possible, what is the likelihood of H246 vs. S95 acting as the nucleophile? The Prezyme calculations suggest that orientation of the substrate favors H246 nucleophile, however, the Mulliken charges suggest that the 5'C carbon, where the catalytic serine has been proposed to attack in AtD14 [1], has a higher partial positive charge. This indicates that the 5'C carbon may be a stronger electrophile than the 2'C carbon.
5. In the reaction scheme shown in Fig. 5h, it is indicated that the part of 4a that is not part of the butenolide dissociates as a leaving group upon nucleophilic attack of the 2'C carbon on the butenolide, forming the expected product. However, in principle, another possibility upon nucleophilic attack of 2'C is a ring opening by dissociation of the 1'O from 2'C. This would result in a H246 modification with a nearly-intact compound 4a with an open butenolide. Has this modification been observed in mass spectrometry data? If so, it would be strong evidence of H246 nucleophilic attack on the 2'C.
6. The Prezyme probabilities (37.9% for S97 nucleophilic attack to C5' vs. 41.6% for H246 nucleophilic attack on C2') seem to suggest that the H246 is more likely to function as the nucleophile than S95. This implies that abolishing the nucleophilic ability of S95 would decrease hydrolytic activity by less than 50%. How can this be reconciled with the observation that S95A is devoid of hydrolytic activity and previous results from de St. Germain et al showing that S97C in RMS3 exhibited weak hydrolytic

activity [2]?

7. The proposed reaction scheme Fig. S18b suggests that weakening the interaction hydrogen bonding interaction between H246 and S95 should increase the nucleophilicity of H246. This is so that the lone pair on the H246 nitrogen is more available to perform the nucleophilic attack rather than act as a hydrogen bond acceptor. If it is feasible to test the hydrolytic activity of S95 mutations on compound 4g, it would provide stronger evidence of the proposed reaction mechanism.

8. The MD simulations were only run for 100ns, which may not be a sufficient amount of time to calculate configurational probabilities of the substrates and catalytic triad accurately. Do the resulting collision probabilities change when additional simulation time is added?

9. In the proposed modes of action in Fig. 6, the authors suggest that non-CLIM forming compounds are unable to induce formation of the closed state of ShHTL7. Since association with MAX2 has been shown to be a vital component of strigolactone signaling, how does this reconcile with the AtD14/OsD3 crystal structure showing that this complex contains the closed form of AtD14 [1]? In particular, Seto et al. suggest that an unhydrolyzed strigolactone can bind to and stabilize the closed form of AtD14 [3]. Is a similar mode of action expected with the reported ShHTL7 agonists?

Minor Point

1. In Fig. 5h, the indicated leaving group (compound 4a without the butenolide ring) appears to be missing a carboxylate group.

References

- [1] Yao, R. et al. DWARF14 is a non-canonical hormone receptor for strigolactone. *Nature* 536, 469–473 (2016).
- [2] de Saint Germain, A. et al. An histidine covalent receptor and butenolide complex mediates strigolactone perception. *Nature Chemical Biology* 12, 787–794 (2016).
- [3] Seto, Y. et al. Strigolactone perception and deactivation by a hydrolase receptor DWARF14. *Nature Communications* 10 (2019).

Reviewer #2 (Remarks to the Author):

In this paper, the authors present novel small molecule suicidal germination inducers for seeds of the parasitic plants *Striga hermonthica* and *Phelipanche aegyptiaca*. While the overall story is interesting, certain aspects should be improved:

1. In line 154, it is stated that the compounds display competitive inhibition of ShHTL7. The mode of inhibition is hard to tell from the data provided. If the authors wish to claim that these compounds are competitive enzyme inhibitors, they could demonstrate this in e.g. a Lineweaver-Burk plot.

2. There are 5 KAI2 protein homologs in *Phelipanche aegyptiaca* and about 10 HTL proteins in *Striga hermonthica*, all of which are potential targets of these germination inducers. These proteins (and not only one of them) should be tested for interaction with the small molecule agonists in either biochemical assays, Y2H, or cross-species complementation experiments.

3. A major claim of the manuscript seems to be the discovery of a new ligand binding mechanism that involves a nucleophilic attack by the NE2 atom the active site histidine. Since a histidine is usually (if at all) a rather weak nucleophile, this is an interesting theory, for which, unfortunately no direct evidence is provided. The only experimental data addressing this is the covalent adduct to the histidine, which does not directly prove that the nucleophilic attack comes from the nitrogen atom.

Minor points:

Line 70: "Burger et al, using modelling". It is unclear what modelling means in this context. Carlsson et al. (2018) J Exp Bot and Burger and Chory (2020) Trends Plant Sci both used the deposited X-ray data from the PDB, refined it with CLIM, found that there was a lot of negative electron density that wouldn't support the presence of the CLIM atoms in the crystal, and then refined the same X-ray data with a different ligand that each provided a better fit. The term modelling does not seem to be an appropriate description of that process.

Throughout the text, the authors should replace GR24 with a clear description of what was used, e.g. rac-GR24, or (+)-GR24...

Line 170: "The YLG assay was performed with ShHTL7, not the receptor protein from *P. aegyptiaca*". This suggests that *P. aegyptiaca* has only 1 receptor protein and should be rephrased.

Reviewer #3 (Remarks to the Author):

This publication presents very relevant results in the field of crop protection against some parasitic weeds like witchweeds and broomrapes. There's been a lot of research on this field on the last decades looking for active germinating stimulants that can be used for the suicidal germination approach to try to save billions of dollars a year.

Xi et al. present a series of potential compound candidates (small-molecule agonists that stimulate *Phelipanche aegyptiaca* and *Striga hermonthica*) for solving this problem with an activity comparable (or even better) than the one of GR24 (the reference compound in the field). The authors also present their study and advances on the elucidation of the mechanism of action of these compounds, which could accelerate the progress on the synthesis of new active molecules to eliminate the parasitic weeds. Their research showed that full efficiency of the synthetic SL agonists in triggering signaling through the *Striga* SL receptor (ShHTL7) depends on the receptor catalyzed hydrolytic reaction of the agonists. They also proposed a new hydrolytic mechanism, although it should be considered just a proposal because more evidence is required to prove it.

The work presented in this publication supports efficiently most of the conclusions and claims drawn at the end of the study and meets the expected standards in the field. All the results are presented with enough detail to be reproduced by an expert in the field.

The authors should make major revisions on the article (see doc file) before it could be published.

Point-by-point responses to Reviewers' Comments

We highly appreciate the reviewers' time and constructive suggestions. These comments are very valuable for improving the quality of this manuscript. The manuscript has been carefully revised according to the reviewers' advice, as detailed below, and the amendments are highlighted in red in the revised manuscript and supporting information.

Reviewer #1 (Remarks to the Author):

This manuscript by Wang et al. presents a set of rationally designed agonists for ShHTL7 that are able to act as germination stimulants for broomrape and witchweed. Using these compounds, the authors also suggest that full signaling activity of the receptors depends on substrate hydrolysis. Additionally, given the difference in bioactivity toward ShHTL7 vs. AtD14, the reported 4a compound appears to be a promising candidate for selective agonism for parasitic weeds. The authors also present evidence that strigolactone hydrolysis may occur via an SN₂-like mechanism involving nucleophilic attack of H247 on the butenolide ring. While this is a very comprehensive and well-written manuscript overall, I have several suggestions for improvements. In particular, while the presented results (particularly on compound 4g) do indicate that the newly proposed reaction mechanism is possible, it is not convincing that this reaction would occur substantially in strigolactones or strigolactone analogs containing an intact butenolide ring.

Major Points

Q1: On pg. 6, line 126, the authors point out that “compounds with electron-withdrawing groups on the benzoyl moiety had a smaller K_D than those with electron-donating groups.” Since energetic contributions of substrate hydrolysis are often not distinguishable from energetic contributions of substrate binding in in vitro K_D measurements, could this observation be explained by change in reactivity of the substrate?

A: The binding free energy (ΔG_{bind}) of a molecule to an enzyme can be calculated using the Arrhenius equation: $\Delta G_{\text{bind}} = -RT \ln K_D$ (Cournia et al. *J. Chem. Inf. Model.* **2017**, *57*, 2911–2937.). If the enzyme-catalyzed reaction follows the Michaelis-Menten two-step model: binding of substrate, followed by catalyzed conversion of substrate to product. The change of the standard free energy (ΔG°) can be calculated using the equation: $\Delta G^\circ = -RT \ln K_m$, where $K_m \approx K_d$ (Silverstein T. P. *Biophys. Chem.* **2021**, *274*, 106606.). The compounds (e. g. **2e** and **2g**) discussed in this sentence could not be hydrolyzed by ShHTL7, so they are not substrates of the enzyme, and their free energies can not be calculated. In the K_D assay, we evaluated the binding affinity between molecule and ShHTL7, and no substrate was added. The obtained K_D values did not seem to have a relationship with the reactivity

of the substrate. Therefore, we mainly discussed the electronic effect of substituents on binding affinities (K_D) of molecules to ShHTL7.

Q2: On pg. 11, line 227-229, the authors state that the CLIM molecule was detected by Q-TOF mass spectrometry. The CLIM molecule is an open form of the butenolide ring that is covalently linked to H246 and S95 and thus would have the same mass as a butenolide ring. Is there additional data that attributes this to a CLIM molecule rather than a butenolide ring attached to H246 or S95?

A: The CLIM molecule formed between H246 and S95 is highly unstable, and will quickly transform to the butenolide ring attached to H246. In the MS-MS analysis, we observed the butenolide ring attached to H246, and not S95. We have revised this sentence as: “, would result in a *covalently-linked molecule in ShHTL7 by using Q-TOF mass spectrometry*” (line 227 in the revised manuscript).

Q3: On pg. 11, line 239-240, the authors state that they assume that the absence of CLIM is “due to the fact that the N-lactam modified His246 is unstable.” This is a reasonable explanation, however, another contributing factor could be decreased electrophilicity of the adjoining 2'C and 5'C carbons vs. when oxygen is present in the same position on the substrate. Could the authors comment on this possibility?

A: Thank you for the comment. We agree with the reviewer that substituting the 1'O with N-CH₃ could decrease the electrophilicity of the adjoining 2'C and 5'C carbons, due to the electron-donating effect of N-CH₃. As shown in Supplementary Fig. 11 (line 145 in the Supplementary Information), ShHTL7 could hydrolyze compound **4h**. This result suggests that though the electrophilicity of the adjoining 2'C and 5'C carbons of **4h** were decreased, it still could be attacked by the nucleophilic residues of ShHTL7. In line 206–208 of the revised manuscript, we have added, “*This is due to the higher electron-donating effect of N-CH₃ of the D-lactam functional group of 4h compared to the oxygen of the D-ring of 4a, facilitating the detachment of the D-lactam from the rest of the molecule.*”, and a similar effect could also facilitate the detachment of the D-lactam from His246 more quickly.

Q4: While hydrolytic activity on compound **4g** does suggest that the newly reported mechanism is possible, what is the likelihood of H246 vs. S95 acting as the nucleophile? The Prezyme calculations suggest that orientation of the substrate favors H246 nucleophile, however, the Mulliken charges suggest that the 5'C carbon, where the catalytic serine has been proposed to attack in AtD14 [1], has a higher partial positive charge. This indicates that the 5'C carbon may be a stronger electrophile than the 2'C carbon.

A: Although the 5'C of **4g** has a higher partial positive charge, hydrolysis through nucleophilic attack of S95 to the 5'C carbon can not result in a covalently-linked modification of H246 of ShHTL7

(Supplementary Fig. 19c). Accordingly, the hydrolytic reaction would happen through the nucleophilic attack of H246 to the 2'C carbon. Indeed, we showed that **4g** can be hydrolyzed by ShHTL7^{S95A} (Supplementary Fig. 24a). We also found a molecule with a mass of 95 Da, corresponding with the modified D-ring, covalently linked to ShHTL7^{S95A} (Supplementary Fig. 24e). Together, this demonstrates that H246 can act as a nucleophile.

Supplementary Fig. 24. **a**, Relative percentage of hydrolysis of compound **4g** by ShHTL7^{S95A}, ShHTL7^{S95C}, and ShHTL7. **b**, LC-MS analysis of the full-length His-tagged ShHTL7 in native condition. **c**, LC-MS analysis of the full-length His-tagged ShHTL7^{S95A} in native condition, the mass of 34699 Da (reduction of 15 Da compared with that of ShHTL7) is the peak of ShHTL7^{S95A}. **d**, LC-MS analysis of full-length His-tagged ShHTL7^{S95C} in native condition, the mass increment of 18 Da compared with that of ShHTL7 suggesting that ShHTL7^{S95C} is in reduction state. **e**, LC-MS analysis of the **4g**-ShHTL7^{S95A} reaction system in the native condition, an increase of *m/z* by 94 Da indicating that the modified D-ring of **4g** is covalently linked to the ShHTL7^{S95A}. **f**, LC-MS analysis of the **4g**-ShHTL7^{S95C} reaction system in the native condition, an increase of *m/z* by 96 Da indicating that the modified D-ring of **4g** is covalently linked to the ShHTL7^{S95C}.

Q5: In the reaction scheme shown in Fig. 5h, it is indicated that the part of **4a** that is not part of the butenolide dissociates as a leaving group upon nucleophilic attack of the 2'C carbon on the butenolide, forming the expected product. However, in principle, another possibility upon nucleophilic attack of 2'C is a ring opening by dissociation of the 1'O⁺ from 2'C. This would result in a H246 modification with a nearly-intact compound **4a** with an open butenolide. Has this modification been observed in mass spectrometry data? If so, it would be strong evidence of H246 nucleophilic attack on the 2'C.

A: We have carefully checked the LC-MS spectrum of the (*S*)-**4a**-ShHTL7 reaction system and did not find the mass peak of ShHTL7+**4a** (around 31091, Fig. R1a). A possible reason for this may be that the intermediate of H246 modified with a nearly-intact **4a** with an open butenolide is highly unstable. It will quickly transform into a more stable form (Fig. R1b). Additionally, after carefully

checking the simulated binding mode of the (*S*)-**4a**–ShHTL7 system, we found that the leaving group (compound **4a** without the butenolide ring) and the imidazole ring of His246 were located at the opposite side of the D-ring (Fig. R1c). As the distance of the hydrogen bond length for H246 and D217 was around 2.0 Å (Supplementary Fig. 22), this suggests that there exist low-barrier hydrogen bonds between H246 and D217, which, in turn, will increase the electronic cloud density of NE2 atom of H246 (Dai et al. *Nature*, **2019**, 573, 609–613; *Science*, **1994**, 264, 1887–1890.). Therefore, we propose that the nucleophilic attack of 2'C by H246 is an SN2 mechanism (Fig. R1d). Additionally, the hydrogen-bonding interaction between the leaving group and S154 and Y174 could also activate this leaving group (Fig. 5h).

Fig. R1. **a**, The LC-MS analysis of the full-length protein of (*S*)-**4a**–ShHTL7 reaction system. **b**, A proposed schematic diagram of ShHTL7-mediated (*S*)-**4a** hydrolysis pathway. In this pathway, the reaction was initiated by the nucleophilic attack of H246 to the 2'C, and the D-ring is opening by dissociating the 1'O from 2'C to form the highly unstable intermediate. The intermediate could fast transform to a more stable format. **c**, The simulated binding mode of (*S*)-**4a** to ShHTL7. **d**, A proposed alternative schematic diagram of ShHTL7-mediated (*S*)-**4a** hydrolysis pathway. In this pathway, His246 attacks to the 2'C through the SN2 mechanism, and the hydrogen-bonding interactions of the leaving group of (*S*)-**4a** with Y174 and Y154 could promote the SN2 reaction.

Q6: The Prezyme probabilities (37.9% for S97 nucleophilic attack to C5' vs. 41.6% for H246 nucleophilic attack on C2') seem to suggest that the H246 is more likely to function as the nucleophile than S95. This implies that abolishing the nucleophilic ability of S95 would decrease hydrolytic activity by less than 50%. How can this be reconciled with the observation that S95A is devoid of hydrolytic activity and previous results from de St. Germain et al showing that S97C in RMS3 exhibited weak hydrolytic activity [2]?

A: Our Prezyme method is based on the statistics of the hydrogen bond length of D217–H246 and H246-S95, and the distance of 5'C-OG or 2'C-NE2 during the molecular dynamic simulation. The

hydrogen bond length between D217 and H246 was around 2.0 Å during the 1000 ns molecular dynamic simulation (Supplementary Fig. 22), suggesting that the hydrogen bond length between D217 and H246 is suitable for enzymatic catalysis. Therefore, the hydrogen bond distance between H246 and S95 is important for Prenzyme calculation. S95A mutation would lead to the disappearance of the hydrogen bond between A95 and H246 (Pang et al. *J. Agric. Food Chem.* **2020**, *68*, 12729–12737.). Accordingly, the Prenzyme probabilities of S95A were zero. Similarly, S96C in RMS3 would weaken the hydrogen-bonding interaction between S96C and His247, decreasing the Prenzyme probability.

Q7: The proposed reaction scheme Fig. S18b suggests that weakening the interaction hydrogen bonding interaction between H246 and S95 should increase the nucleophilicity of H246. This is so that the lone pair on the H246 nitrogen is more available to perform the nucleophilic attack rather than act as a hydrogen bond acceptor. If it is feasible to test the hydrolytic activity of S95 mutations on compound **4g**, it would provide stronger evidence of the proposed reaction mechanism.

A: We have tested the hydrolytic activity of ShHTL7^{S95A} and ShHTL7^{S95C} mutations on compound **4g**. The results show that S95A and S95C mutations of ShHTL7 decrease the hydrolytic activity towards **4g** (Supplementary Fig. 24a). To investigate whether the hydrolysis of **4g** by ShHTL7^{S95A} and ShHTL7^{S95C} results in covalent modification of the proteins, we performed LC-MS analysis of **4g**-ShHTL7^{S95A} and **4g**-ShHTL7^{S95C} reaction systems (Supplementary Fig. 24b-f). An increase by 94 Da was detected in **4g** treated ShHTL7^{S95A} protein (Supplementary Fig. 24e), which corresponds to the modified D-ring of **4g** covalently bound to the ShHTL7^{S95A} protein. Similarly, an increase by 96 Da was detected in **4g** treated ShHTL7^{S95C} protein (Supplementary Fig. 24f), which corresponds to the modified D-ring of **4g** covalently bound to the ShHTL7^{S95C} protein. These results show that the nucleophilic attack of H246 on the 2'C is possible. The new Supplementary Fig. 24 was added to line 234–244 of the revised Supplementary Information. In line 351–354 of the revised manuscript, we have added: “*We also observed that 4g could be hydrolyzed by ShHTL7^{S95A} and ShHTL7^{S95C} mutant proteins (Supplementary Fig. 24a). The LC-MS analysis showed that a molecule with m/z around 95, which corresponds to the modified D-ring of 4g, is covalently linked to the ShHTL7^{S95A} and ShHTL7^{S95C} proteins (Supplementary Fig. 24b-f).*” In line 397-398 of the revised manuscript, we have added: “*, MS-MS analysis and S95 mutations studies.*”.

Supplementary Fig. 24. **a**, Relative percentage of hydrolysis of compound **4g** by ShHTL7^{S95A}, ShHTL7^{S95C}, and ShHTL7. **b**, LC-MS analysis of the full-length His-tagged ShHTL7 in native condition. **c**, LC-MS analysis of the full-length His-tagged ShHTL7^{S95A} in native condition, the mass of 34699 Da (reduction of 15 Da compared with that of ShHTL7) is the peak of ShHTL7^{S95A}. **d**, LC-MS analysis of full-length His-tagged ShHTL7^{S95C} in native condition, the mass increment of 18 Da compared with that of ShHTL7 suggesting that ShHTL7^{S95C} is in reduction state. **e**, LC-MS analysis of the **4g**–ShHTL7^{S95A} reaction system in the native condition, an increase of m/z by 94 Da indicating that the modified D-ring of **4g** is covalently linked to the ShHTL7^{S95A}. **f**, LC-MS analysis of the **4g**–ShHTL7^{S95C} reaction system in the native condition, an increase of m/z by 96 Da indicating that the modified D-ring of **4g** is covalently linked to the ShHTL7^{S95C}.

In addition, we have provided the following evidence to support the role of H246 as a nucleophile:

1. In the enzyme catalytic reactions, histidines are catalytically versatile residues, not only because the imidazole function can serve as both hydrogen bond acceptor and donor, but also due to the nucleophilicity of the NE2 atom (Röthlisberger et al. *Nature*, **2008**, *453*, 190–195; Burke et al. *Nature*, 2019, *570*, 219–223.). For example, Shah *et al.* reported that the NE2 atom of His112 of hHint1 could attack the phosphorus of the substrate (*Biochemistry* **2017**, *56*, 3559–3570.), and similar nucleophilic cases can also be found in the phospholipase D and the histidine triad superfamilies (Selvy, *et al. Chem. Rev.* **2011**, *111*, 6064–6119; Pluta *et al. Proc. Natl. Acad. Sci.* **2017**, *114*, E6526–E6535. Casino *et al. Nat. Commun.* **2014**, *5*, 3258.). Nie *et al.* reported that His150 of the ELOVL elongases serve as a nucleophile in the enzyme catalytic process, and the nucleophilicity of His150 could be enhanced by the hydrogen bonding integrations with other residues (*Nat. Struct. Mol. Biol.* **2021**, *28*, 512–520.). Morgen *et al. (ACS Chem. Biol.* **2016**, *11*, 1001–1011.) reported that the NE2 atom of His231 of MetAP2 could attack the sp² carbon of fumagillin, resulting in the irreversible inhibition of the protein (Gehringer et al. *J. Med. Chem.* **2019**, *62*, 5673–5724.). Kuroki et al. reported

that the His26 of the T26H reengineered T4 lysozyme could act as a nucleophile during the catalytic process (*Proc. Natl. Acad. Sci.* **1999**, *96*, 8949-8954.). Burke *et al.* also reported that the artificial hydrolytic enzyme could use histidine as a nucleophile to form acyl-enzyme intermediate (*Nature*, **2019**, *570*, 219–223.).

2. Our Prenzyme calculations suggest that the NE2 atom of H246 can attack the 2'C of the D-ring. Additionally, the hydrogen-bonding interactions of the 2O and 3O atoms of (*S*)-**4a** could form hydrogen-bonding interactions with S154 and Y174, respectively, which, in turn, could activate the leaving group (R₂NC(=O)O⁻) and promote the SN2 reaction.

3. As shown in Fig 3c and Supplementary Fig. 19b-c, the covalent modification of compound **4g** on His246 of ShHTL7 indicates that the newly proposed reaction mechanism is possible.

4. We demonstrate that the nucleophilic attack on the 2'C atom of **4g** by the imidazole group of *N*-acetyl-L-histidine methyl ester could occur under mild reaction conditions (Supplementary Table 5). An enzymatic catalytic reaction is far more efficient than a chemical reaction under normal conditions (Agarwal P. K. *Biochemistry* **2019**, *58*, 438–449.). Therefore, it can be inferred that it would be more efficient when the His246 of ShHTL7 reacts with **4g**, than that under normal conditions, which is also consistent with the results of our experiments.

Supplementary Table 5. The nucleophilic substitution reaction conditions of As-his-ome to the 2'C atom of **4g**.^a

entry	additive (equiv)	solvent	time (h)	temperature (°C)	yield (%) ^b
1	NaH (1.2)	THF	6	0 to 25	trace
2	K ₂ CO ₃ (1.25)	DMF	6	25	trace
3	DBU (1.5)	THF	6	25	trace

^aReactions were performed on a 0.033 mmol scale of **4g** and 10 mL of solvent under the given conditions. ^bThe yields were determined by UPLC-HRMS.

Q8: The MD simulations were only run for 100ns, which may not be a sufficient amount of time to calculate configurational probabilities of the substrates and catalytic triad accurately. Do the resulting collision probabilities change when additional simulation time is added?

A: The time of MD simulations was prolonged to 1000 ns (Supplementary Fig. 21). This resulted in minor changes in the configurational probabilities, with the final *P*_{PC} for (*R*)-**4a** (Conf1), (*R*)-**4a** (Conf2), and (*S*)-**4a** being 1.3%, 0%, and 33%, respectively (line 331–332 in the revised manuscript).

Additionally, the P_{PC} of D₂'C-NE₂ and D_{NE2}-HG was calculated to be 19.2% (line 341 in the revised manuscript).

Q9: In the proposed modes of action in Fig. 6, the authors suggest that non-CLIM forming compounds are unable to induce formation of the closed state of ShHTL7. Since association with MAX2 has been shown to be a vital component of strigolactone signaling, how does this reconcile with the AtD14/OsD3 crystal structure showing that this complex contains the closed form of AtD14 [1]? In particular, Seto et al. suggest that an unhydrolyzed strigolactone can bind to and stabilize the closed form of AtD14 [3]. Is a similar mode of action expected with the reported ShHTL7 agonists?

A: Our yeast two-hybrid assays showed that even at a concentration of 200 μ M, **4i** still could not induce the interaction of ShHTL7 with MAX2. GR24 could be hydrolyzed by AtD14 and cause the conformation change of AtD14, which, in turn, induces the association of AtD14 with MAX2 (Yao, et al. *Nature* **2016**, 536, 469–473). This result also suggests that the closed conformation AtD14 is crucial for the interaction of AtD14 with MAX2 and signal transduction (Waters et al. *Annu. Rev. Plant Biol.* **2017**, 68, 291–322; Blázquez et al. *Annu. Rev. Plant Biol.* **2020**, 71, 327-353.). Seto et al. (*Nat. Commun.* **2019**, 10, 191.) also agree that the closed conformation of AtD14 could induce the association of AtD14 with MAX2. In their study, they mainly used the AtD14^{D218A} mutant as a model to conclude that an intact strigolactone molecule triggers AtD14 signaling. After analyzing the crystal structure of AtD14 (PDB id: 4IH4), we find that D218 and H247 are located at two different loops, and the strong hydrogen-bonding interaction between D218 and H217 could stabilize the loops and catalytic triad (Fig. R2a). AtD14^{D218A} could significantly impair the interaction of D218 and H246, making the conformation flexibility of the two loops, which, in turn, would eventually induce the conformation change of the target protein. After comparing the structures of AtD14 proteins in closed (AtD14/OsD3, PDB id: 5HZG) and open forms (Fig. R2b-c). It was observed that there was about 9.2 Å movement of D218 (Fig. R2c); at such a long distance, the hydrogen-bonding interaction between D218 and H217 would disappear. This result also suggests that the AtD14^{D218A} mutation could affect the conformation of the protein. Although AtD14^{D218A} could not hydrolyze strigolactone, once strigolactone binds to the protein, the conformation of AtD14^{D218A} could more easily stabilize to the closed-form and recruit the downstream proteins. Based on these considerations, we can infer that at the open conformation state, AtD14 could not have an association with MAX2. Therefore, we proposed that: “For hydrolysis-resistant agonists (i.e. compound **4i**), ShHTL7 might bind with the intact molecule in the open conformation and trigger downstream signaling; however, the efficiency is rather low.” (line 769–771 in the revised manuscript).

Fig. R2. a, The crystal structure of the AtD14 protein (PDB id: 4IH4). The protein is shown in green cartoon, and the amino acids are presented by sticks. **b**, The crystal structure of AtD14/OsD3 complex (PDB id: 5HZG). AtD14 is shown in magenta cartoon, D3 LRR is shown by wheat surface, and ASK1 is shown by cyan surface. **c**, Alignment of AtD14 in open (green cartoon) and closed (magenta cartoon) states.

Minor Point

Q10: In Fig. 5h, the indicated leaving group (compound **4a** without the butenolide ring) appears to be missing a carboxylate group.

A: We have detected the leaving group (compound (*S*)-**4a** without the butenolide ring, designated **L1**) by analyzing the mass spectrum of the (*S*)-**4a** and ShHTL7 reaction system, and a compound with a mass of 313.0566 corresponds well to the mass of 2-(4-bromo-2-methylphenoxy)-1-(piperazin-1-yl)ethan-1-one, HRMS (ESI): calcd for $C_{13}H_{18}BrN_2O_2$ $[M+H]^+$: 313.0552 (Supplementary Fig. 23a). Whereas the mass of the leaving group with a carboxylate group (designated **L2**) was not detected, suggesting that **L2** is highly unstable and will immediately transform to **L1** by releasing CO_2 , which, in turn, could facilitate the enzymatic hydrolysis of (*S*)-**4a** (Supplementary Fig. 23b). The new Supplementary Fig. 23 was added to line 227–233 of the revised Supplementary Information. In line 345 of the revised manuscript, we have added: “(Fig. 5h, *Supplementary Fig. 23*).”

Supplementary Fig. 23. **a**. The mass spectrum of the leaving group (compound (*S*)-**4a** without the butenolide ring, designated **L1**) in the (*S*)-**4a** and ShHTL7 reaction system. **b**. A proposed pathway for the formation of **L1** in the (*S*)-**4a** and ShHTL7 reaction system. After incubating (*S*)-**4a** with ShHTL7, the D-ring was covalently linked to the His246 and released the highly unstable intermediate **L2**. The **L2** would immediately degrade to **L1** by releasing one molecule of CO_2 , which, in turn, could promote the enzymatic catalytic reaction.

References

- [1] Yao, R. et al. DWARF14 is a non-canonical hormone receptor for strigolactone. *Nature* 536, 469–473 (2016).
- [2] de Saint Germain, A. et al. An histidine covalent receptor and butenolide complex mediates strigolactone perception. *Nature Chemical Biology* 12, 787–794 (2016).
- [3] Seto, Y. et al. Strigolactone perception and deactivation by a hydrolase receptor DWARF14. *Nature Communications* 10 (2019).

Reviewer #2 (Remarks to the Author):

In this paper, the authors present novel small molecule suicidal germination inducers for seeds of the parasitic plants *Striga hermonthica* and *Phelipanche aegyptiaca*. While the overall story is interesting, certain aspects should be improved:

Q1: In line 154, it is stated that the compounds display competitive inhibition of ShHTL7. The mode of inhibition is hard to tell from the data provided. If the authors wish to claim that these compounds are competitive enzyme inhibitors, they could demonstrate this in e.g. a Lineweaver-Burk plot.

A: We have evaluated inhibitory kinetics of the three representative compounds **1**, **2g**, and **2h** on ShHTL7-mediated hydrolysis of YLG. The results showed that no significant change of the maximal velocity (V_{max}) was observed in the Lineweaver–Burk plot (Supplementary Fig. 7), indicating that **1**, **2g**, and **2h** inhibit the hydrolytic activity of ShHTL7 in a competitive manner. The new Supplementary Fig.7 was added to line 126–128 of the revised Supplementary Information. The assay methods for the Lineweaver–Burk plots were shown in line 72–82 of the revised Supplementary Information. In line 154 of the revised manuscript, we have added: “(Supplementary Fig.7, Supplementary Table 1)”.

Supplementary Fig. 7. Lineweaver–Burk plots for the inhibition of ShHTL7 by compounds **1** (a), **2g** (b), and **2h** (c). Error bar indicates SD ($n = 3$ biological replicates).

Q2: There are 5 KAI2 protein homologs in *Phelipanche aegyptiaca* and about 10 HTL proteins in

Striga hermonthica, all of which are potential targets of these germination inducers. These proteins (and not only one of them) should be tested for interaction with the small molecule agonists in either biochemical assays, Y2H, or cross-species complementation experiments.

A: Thank you for the suggestion. We have evaluated the competitive inhibitory activity of compounds **4a**, (*R*)- and (*S*)-**4a**, and **4g-i** to the 10 HTL proteins in *S. hermonthica* and 5 KAI2 protein homologs in *P. aegyptiaca* using YLG as the fluorogenic substrate (Supplementary Fig. 27). The new Supplementary Fig. 27 was added to line 256–260 of the revised Supplementary Information. In line 408–411 of the revised manuscript, we have added: “Additionally, the YLG-based competition assay for the 10 HTL proteins in *S. hermonthica* and 5 KAI2s protein homologs in *P. aegyptiaca* also suggest that **4a** and (*S*)-**4a** bind to these proteins, with IC_{50} values ranging from 0.94 to 55.76 μ M (Supplementary Fig. 27).”

	4a	(R)-4a	(S)-4a	4g	4h	4i	rac -GR24
ShHTL2	55.76 ±6.58	66.08 ±3.16	47.42 ±3.02	65.24 ±0.72	76.57 ±3.74	61.94 ±10.13	25.65 ±0.52
ShHTL3	43.03 ±1.58	67.32 ±8.14	15.65 ±1.09	34.07 ±6.03	>100	>100	8.40 ±1.17
ShHTL4	4.95 ±1.18	20.99 ±1.66	2.56 ±0.37	17.71 ±1.59	12.09 ±2.86	17.52 ±3.27	0.25 ±0.02
ShHTL5	15.69 ±2.70	43.91 ±6.97	5.88 ±0.82	39.18 ±3.15	38.11 ±4.74	44.70 ±3.69	6.49 ±0.53
ShHTL6	23.46 ±4.99	>100	8.05 ±1.69	50.63 ±12.21	>100	67.88 ±12.22	2.13 ±0.22
ShHTL7	1.21 ±0.22	35.80 ±3.69	0.94 ±0.11	2.22 ±0.77	4.84 ±0.56	5.18 ±0.81	0.70 ±0.13
ShHTL8	17.15 ±3.79	60.52 ±8.50	8.54 ±3.15	58.27 ±5.60	75.82 ±8.61	73.09 ±2.17	5.81 ±1.17
ShHTL9	22.93 ±3.25	47.31 ±3.17	15.39 ±4.11	38.71 ±7.52	>100	20.06 ±6.44	4.71 ±0.58
ShHTL10	33.93 ±4.96	52.54 ±8.25	12.03 ±0.66	53.14 ±3.09	43.60 ±3.24	64.99 ±6.27	5.67 ±0.83
ShHTL11	10.73 ±2.09	>100	5.33 ±1.40	56.25 ±1.78	>100	93.04 ±7.74	3.19 ±0.07
Phelipanche_aegyptiaca_KAI2c	15.38 ±5.48	51.37 ±6.02	5.57 ±0.43	48.47 ±5.19	32.90 ±8.35	51.74 ±4.59	6.52 ±0.81
Phelipanche_aegyptiaca_KAI2d1	30.47 ±4.94	>100	13.62 ±1.63	24.90 ±1.97	24.88 ±2.90	20.27 ±3.64	2.33 ±0.53
Phelipanche_aegyptiaca_KAI2d2	45.34 ±4.09	>100	37.77 ±3.44	69.98 ±6.72	70.13 ±2.42	56.45 ±1.96	3.83 ±1.17
Phelipanche_aegyptiaca_KAI2d3	28.22 ±2.12	84.90 ±6.50	9.45 ±2.13	49.19 ±11.07	37.91 ±6.32	44.27 ±6.15	1.95 ±0.37
Phelipanche_aegyptiaca_KAI2d4	20.47 ±5.08	>100	6.92 ±1.33	79.56 ±5.00	>100	29.77 ±2.52	6.50 ±0.71

IC_{50} (μ M)

Supplementary Fig. 27. Binding of compounds **4a**, (*R*)- and (*S*)-**4a**, and **4g-i** to the HTL proteins in *S. hermonthica* and KAI2 protein homologs in *P. aegyptiaca*. In the competitive inhibitory activity assay, YLG (3 μ M) was used as the fluorogenic substrate. Error bar indicates SD (n = 3 biological replicates).

Q3: A major claim of the manuscript seems to be the discovery of a new ligand binding mechanism that involves a nucleophilic attack by the NE2 atom the active site histidine. Since a histidine is usually (if at all) a rather weak nucleophile, this is an interesting theory, for which, unfortunately no direct evidence is provided. The only experimental data addressing this is the covalent adduct to the histidine, which does not directly prove that the nucleophilic attack comes from the nitrogen atom.

A: Histidine residues are not commonly reported as nucleophiles in the enzyme catalytic process, but there are studies showing this. In this manuscript, we proposed nucleophilic attack of the NE2 atom of H246 to the 2'C of the D-ring, based on the following considerations:

1. In the enzyme catalytic reactions, histidines are catalytically versatile residues, not only because the imidazole function can serve as both hydrogen bond acceptor and donor, but also due to the nucleophilicity of the NE2 atom (Röthlisberger et al. *Nature*, **2008**, 453, 190–195; Burke et al. *Nature*, 2019, 570, 219–223.). For example, Shah *et al.* reported that the NE2 atom of His112 of hHint1 can attack the phosphorus of the substrate (*Biochemistry* **2017**, 56, 3559–3570.), and similar nucleophilic cases can also be found in phospholipase D and the histidine triad superfamilies (Selvy, *et al. Chem. Rev.* **2011**, 111, 6064–6119; Pluta *et al. Proc. Natl. Acad. Sci.* **2017**, 114, E6526-E6535. Casino *et al. Nat. Commun.* **2014**, 5, 3258.). Nie *et al.* reported that His150 of the ELOVL elongases serves as a nucleophile in the enzyme catalytic process, and that the nucleophilicity of His150 can be enhanced by hydrogen bonding with other residues (*Nat. Struct. Mol. Biol.* **2021**, 28, 512-520.). Morgen *et al. (ACS Chem. Biol.* **2016**, 11, 1001–1011.) reported that the NE2 atom of His231 of MetAP2 can attack the sp² carbon of fumagillin, resulting in the irreversible inhibition of the protein (Gehring et al. *J. Med. Chem.* **2019**, 62, 5673–5724.). Kuroki et al. reported that the His26 of the T26H reengineered T4 lysozyme could act as a nucleophile during the catalytic process (*Proc. Natl. Acad. Sci.* **1999**, 96, 8949-8954.). Burke *et al.* also reported that an artificial hydrolytic enzyme could use histidine as a nucleophile to form an acyl-enzyme intermediate (*Nature*, **2019**, 570, 219–223.). In line 200–201 of the revised Supplementary Information, we have added: *Similar examples of NE2 atoms of histidines acting as nucleophiles can also be found in the literature*^{13–22}.

2. Our Prenzyme calculation suggested that the NE2 atom of H246 could attack the 2'C of the D-ring. Additionally, the hydrogen-bonding interactions of the 2O and 3O atoms of (*S*)-**4a** could form hydrogen-bonding interactions with S154 and Y174, respectively, these interactions which, in turn, could activate the leaving group (R₂NC(=O)O⁻) and promote the SN₂ reaction.

3. As shown in Fig 3c and Supplementary Fig. 19b-c, the covalent modification of compound **4g** on His246 of ShHTL7 indicates that the newly proposed reaction mechanism is possible.

4. We have found that the nucleophilic attack on the 2'C atom of **4g** by the imidazole group of *N*-acetyl-L-histidine methyl ester could occur under the mild reaction conditions (Supplementary Table 5).

Supplementary Table 5. The nucleophilic substitution reaction conditions of As-his-ome to the 2'C atom of **4g**.^a

entry	additive (equiv)	solvent	time (h)	temperature (°C)	yield (%) ^b
1	NaH (1.2)	THF	6	0 to 25	trace

2	K ₂ CO ₃ (1.25)	DMF	6	25	trace
3	DBU (1.5)	THF	6	25	trace

^aReactions were performed on a 0.033 mmol scale of **4g** and 10 mL of solvent under the given conditions. ^bThe yields were determined by UPLC-HRMS.

5. We have also found that the ShHTL7^{S95A} and ShHTL7^{S95C} mutations could hydrolyze **4g** (Supplementary Fig. 24a). Our LC-MS analysis of **4g**-ShHTL7^{S95A} and **4g**-ShHTL7^{S95C} reaction systems showed that after incubating **4g** with ShHTL7^{S95A}, an increase by 94 Da was detected in the **4g** treated ShHTL7^{S95A} protein (Supplementary Fig. 24e), which corresponds to the modified D-ring of **4g** covalently bound to the ShHTL7^{S95A} protein. Similarly, an increase by 96 Da was detected in **4g** treated ShHTL7^{S95C} protein (Supplementary Fig. 24f), which corresponds to the modified D-ring of **4g** covalently bound to the ShHTL7^{S95C} protein. These results showed that the nucleophilic attack of H246 on the 2'C is possible.

Collectively, the above results confirm that the nucleophilic attack of H246 on the 2'C is possible.

Supplementary Fig. 24. **a**, Relative percentage of hydrolysis of compound **4g** by ShHTL7^{S95A}, ShHTL7^{S95C}, and ShHTL7. **b**, LC-MS analysis of the full-length His-tagged ShHTL7 in native condition. **c**, LC-MS analysis of the full-length His-tagged ShHTL7^{S95A} in native condition, the mass of 34699 Da (reduction of 15 Da compared with that of ShHTL7) is the peak of ShHTL7^{S95A}. **d**, LC-MS analysis of full-length His-tagged ShHTL7^{S95C} in native condition, the mass increment of 18 Da compared with that of ShHTL7 suggesting that ShHTL7^{S95C} is in reduction state. **e**, LC-MS analysis of the **4g**-ShHTL7^{S95A} reaction system in the native condition, an increase of *m/z* by 94 Da indicating that the modified D-ring of **4g** covalently linked to the ShHTL7^{S95A}. **f**, LC-MS analysis of the **4g**-ShHTL7^{S95C} reaction system in the native condition, an increase of *m/z* by 96 Da indicating that the modified D-ring of **4g** covalently linked to the ShHTL7^{S95C}.

Minor points:

Q4: Line 70: "Burger et al, using modelling". It is unclear what modelling means in this context. Carlsson et al. (2018) J Exp Bot and Burger and Chory (2020) Trends Plant Sci both used the deposited X-ray data from the PDB, refined it with CLIM, found that there was a lot of negative electron density that wouldn't support the presence of the CLIM atoms in the crystal, and then refined the same X-ray data with a different ligand that each provided a better fit. The term modelling does not seem to be an appropriate description of that process.

A: We have changed this sentence to: “, while *Bürger et al*, refined the X-ray data and proposed that” (line 70 in the revised manuscript).

Q5: Throughout the text, the authors should replace GR24 with a clear description of what was used, e.g. rac-GR24, or (+)-GR24...

A: In this manuscript, we use *rac*-GR24 as the positive control, and GR24 was replaced with *rac*-GR24 throughout the text.

Q6: Line 170: "The YLG assay was performed with ShHTL7, not the receptor protein from *P. aegyptiaca*". This suggests that *P. aegyptiaca* has only 1 receptor protein and should be rephrased.

A: We have rephrased this sentence as “*The YLG assay was performed with ShHTL7, not the protein homologs from P. aegyptiaca*” (line 169–170 in the revised manuscript).

Reviewer #3 (Remarks to the Author):

This publication presents very relevant results in the field of crop protection against some parasitic weeds like witchweeds and broomrapes. There's been a lot of research on this field on the last decades looking for active germinating stimulants that can be used for the suicidal germination approach to try to save billions of dollars a year.

Xi et al. present a series of potential compound candidates (small-molecule agonists that stimulate *Phelipanche aegyptiaca* and *Striga hermonthica*) for solving this problem with an activity comparable (or even better) than the one of GR24 (the reference compound in the field). The authors also present their study and advances on the elucidation of the mechanism of action of these compounds, which could accelerate the progress on the synthesis of new active molecules to eliminate the parasitic weeds. Their research showed that full efficiency of the synthetic SL agonists in triggering signaling through the *Striga* SL receptor (ShHTL7) depends on the receptor catalyzed

hydrolytic reaction of the agonists. They also proposed a new hydrolytic mechanism, although it should be considered just a proposal because more evidence is required to prove it.

The work presented in this publication supports efficiently most of the conclusions and claims drawn at the end of the study and meets the expected standards in the field. All the results are presented with enough detail to be reproduced by an expert in the field.

The authors should make major revisions on the article (see doc file) before it could be published.

Q1: Line 26: “ 10^{-8} to 10^{-17} ”. Minus signs should be used, not hyphens. This should be changed in all the document.

A: We have changed all the hyphens to minus signs.

Q2: Line 31: 2'C, sometimes written 2'C and sometimes 2' C. Be consistent.

A: We have changed “2' C” to “2'C” and made it consistent throughout the manuscript.

Q3: Line 33: “stimulants which” → “stimulants, which”.

A: We have changed “stimulants which” to “stimulants, which” in the revised manuscript (line 33 in the revised manuscript).

Q4: Line 54: Supplementary Fig. 1. It should be specified that the racemates of all the chiral compounds (S5, S6, S7, S8, S10, S11, and S13) were used (except were stated). The same thing for all the other compounds in all the other figures. Besides, the relative configuration compounds with more than one chirality center should be drawn clearly. In compound S8 the H should be drawn on top of the N (the C-N bond cannot be seen the way the molecule is drawn).

A: We have made the above changes in the revised Supplementary Fig. 1 (line 100 in the revised Supplementary Information file).

Q5: Line 54, 55: “2' R configured functional group”. Functional groups don't have R/S configuration, it is the tetrahedral carbon that has the R configuration. Change appropriately.

A: We have changed “2' R configured functional group” to “2'R coupled D-ring” (line 54 in the revised manuscript).

Q6: Line 93: “ μM ” → “micromolar” (use symbols and units only in scientific context, that is, after a number, not in regular text).

A: We have changed “ μM ” to “micromolar” (line 93 in the revised manuscript).

Q7: Line 103: Supplementary Fig. 2 the same as in line 54. Compounds AG-205/36953272, AO-079/14332005, AG-690/36004059, and AA-504/06584039 should be redrawn specifying the relative configuration. Compound AN-584/43408340 should be redrawn in a more clear way (e.g.,

A: We have carefully checked the chiral information of compounds AG-205/36953272, AO-079/14332005, AG-690/36004059, and AA-504/06584039 and consulted the sales manager of the Specs database (www.specs.net). They told us that the relative configuration (chiral info) of these compounds is not available, and that they are probably racemic mixtures. Additionally, we have redrawn the structures of compounds AN-584/43408340 and AO-079/14332005 in a more clear way (line 102 in the revised Supplementary Information file).

Q8: Line 108: “similar as” → “similar to”.

A: We have changed “similar as” to “similar to” (line 108 in the revised manuscript).

Q9: Line 110: “was shown to have” → “had” (write economically).

A: We have changed “was shown to have” to “had” (line 110 in the revised manuscript).

Q10: Line 116: Supplementary Fig. 4 same as line 54.

A: We have also consulted the sales manager of the Specs database about the chiral info of compounds in Supplementary Fig. 4. They told us that the relative configuration (chiral info) of these compounds is not available, and that they are probably racemic mixtures.

Q11: Line 117: delete “Surprisingly” (no human emotions needed, just reporting results).

A: We have deleted “Surprisingly” (line 117 in the revised manuscript).

Q12: Lines 123 and 124: “penta-heterocycles showed higher affinity than “hexa-heterocycles”. This sentence is not clear. Does it mean five-membered heterocycles and six-membered heterocycles? Change accordingly.

A: We have revised this sentence as “...five-membered heterocycles showed higher affinity than six-membered heterocycles...” (line 123–124 in the revised manuscript).

Q13: Line 125: “compared with the”. Does it mean “with respect to”? Not clear.

A: We thank the reviewer for pointing out this potential for confusion. We have changed “compared with the” to “with respect to” (line 124–125 in the revised manuscript).

Q14: Line 124: Molecules 2c, 2g, and 2h in Fig. 1b and Supplementary 4 are different.

A: We apologize that we have made a mistake in drawing the structures of **2c**, **2g**, and **2h** in Supplementary 4. The structures of the three compounds in Fig. 1b were correct, and we have redrawn the structures of them in Supplementary 4 (line 111 in the revised Supplementary 4).

Q15: Line 125: 2f doesn't have a para substituent.

A: We apologize that we did not explain it clearly in this sentence. We want to express that the compound with a para substituent shows higher binding with ShHTL7 (e.g. **2h**) than with the substituent on the other position (e.g. **2f**). We have changed this sentence to “was favorable for binding with ShHTL7 (e.g. **2f** vs **2h**).” (line 125 in the revised manuscript).

Q16: Line 127: compound 2e has an electron-withdrawing group, not an electron-donating group as stated.

A: We thank the reviewer for pointing out this potential for confusion. In this sentence, we want to give an example of compounds with an electron-withdrawing group (e.g. **2e**) and electron-donating group (e.g. **2i**). To make a more clear description, we have revised this sentence to “...compounds with electron-withdrawing groups (e.g. **2e**, Fig. 1b) on the benzoyl moiety had a smaller K_D than those with electron-donating groups (e.g. **2i**, Supplementary Fig. 4)” (line 126–127 in the revised manuscript).

Q17: Line 147: “stereo-isomers” → “stereoisomers”, no hyphen.

A: We have changed “stereo-isomers” to “stereoisomers” (line 146 in the revised manuscript).

Q18: Line 147 and 148: “(3S)-3f” and “(3R)-3f”. The number 3 should not be italicized and it is position 2, not 3 (as indicated in line 122, “bicyclo[2.2.1]hept-5-en-2-yl...”). This should be changed everywhere in the document.

A: We have changed “(3S)-” and “(3R)-” to “(2S)-” and “(2R)-”, respectively, in the whole revised manuscript and Supplementary Information.

Q19: Line 167: “while” → “whereas”.

A: We have changed “while” to “whereas” (line 166 in the revised manuscript).

Q20: Line 172: “1-(bicyclo[2.2.1]hept-5-en-2-ylmethyl)-4-piperazin-1-yl” → “4-(bicyclo[2.2.1]hept-5-en-2-ylmethyl)piperazin-1-yl”.

A: We have changed “1-(bicyclo[2.2.1]hept-5-en-2-ylmethyl)-4-piperazin-1-yl” to “4-(bicyclo[2.2.1]hept-5-en-2-ylmethyl)piperazin-1-yl” (line 173 in the revised manuscript).

Q21: Line 203: “stability” → “reactivity”. Wrong use of the word “stability” (although very spread) with the meaning of “reactivity”.

A: We have changed “stability” to “reactivity” (line 202 in the revised manuscript).

Q22: Line 205: “higher stability” → “lower reactivity”.

A: We have changed “higher stability” to “lower reactivity” (line 204 in the revised manuscript).

Q23: Line 207: “decreased stability” → “higher reactivity”.

A: We have changed “decreased stability” to “higher reactivity” (line 206 in the revised manuscript).

Q24: Lines 207 and 208: “The possible reason is that the N-CH₃ D-lactam functional group of 4h is less stable than the acetal D-ring of 4a”. This comparison is not correct in the sense that it is comparing the amide with the acetal and the only change between 4a and 4h is an ester for an amide. Compound 4h is quite likely more reactive (and almost for sure, more stable) than 4a because of the higher electron-donating effect of the nitrogen compared to the oxygen, facilitating the detachment of the D-ring from the rest of the molecule. Change accordingly.

A: We have changed this sentence to “*This is because of that the higher electron-donating effect of N-CH₃ of D-lactam functional group of 4h compared to the oxygen of the D-ring of 4a, facilitating the detachment of the D-lactam from the rest of the molecule.*” (line 206–208 in the revised manuscript).

Q25: Line 209: “high stability” → “lower reactivity”.

A: We have changed “high stability” to “lower reactivity” (line 209 in the revised manuscript).

Q26: Lines 210 and 211: “chemical stability” → “reactivity”.

A: We have changed “chemical stability” to “reactivity” (line 211 in the revised manuscript).

Q27: Line 211: “stability” → “reactivity”.

A: We have changed “stability” to “reactivity” (line 211 in the revised manuscript).

Q28: Line 212: “stability” → “reactivity”.

A: We have changed “stability” to “reactivity” (line 212 in the revised manuscript).

Q29: Line 213: “stable” → “reactive”.

A: We have changed “stable” to “reactive” (line 214 in the revised manuscript).

Q30: Lines 215 and 216: “This inconsistency is likely explained by the fact that in vitro chemical stability does not equal hydrolytic activity by receptors”. Delete this line. There is no inconsistency: this is a perfectly normal result because the conditions are entirely different: hydrolysis of chiral compounds under achiral conditions compared to hydrolysis under chiral conditions.

A: We have deleted this sentence (line 214–215 in the revised manuscript).

Q31: Line 223: “To our surprise”. Avoid personal feelings reporting results.

A: We have deleted “To our surprise” (line 222 in the revised manuscript).

Q32: Line 240: “very unstable” → “more reactive” (amides are more stable than esters).

A: We have changed “very unstable” to “more reactive” (line 239 in the revised manuscript).

Q33: Line 266: “evaluate this” → “evaluate this result”.

A: We have changed “evaluate this” to “evaluate this result” (line 265 in the revised manuscript).

Q34: Line 269: “This” → “This result”.

A: We have changed “This” to “This result” (line 269 in the revised manuscript).

Q35: Lines 291 and 292: It doesn’t make sense to present the calculation of the Mulliken charge of two enantiomers: it should be the same.

A: In the revised manuscript and Supplementary Information, we only showed the Mulliken charge of (*S*)-**4a** (line 287–290 in the revised manuscript; line 188–194 in the revised Supplementary information).

Q36: Line 293: “to attack by” → “to be attacked by”.

A: We have changed “to attack by” to “to be attacked by” (line 292 in the revised manuscript).

Q37: Lines 294 and 295: “through the nucleophilic attack of the carbonyl group of the D-ring by the Serine residue” → “by the nucleophilic attack of the Serine residue to the carbonyl group of the D-ring”.

A: We have changed “through the nucleophilic attack of the carbonyl group of the D-ring by the Serine residue” to “by the nucleophilic attack of the Serine residue to the carbonyl group of the D-ring” (line 293-294 in the revised manuscript).

Q38: Line 324: “x-ray” → “X-ray”.

A: We have changed “x-ray” to “X-ray” (line 324 in the revised manuscript).

Q39: Line 332: “(Fig. 5g), respectively” → “respectively (Fig. 5g)”.

A: We have changed “(Fig. 5g), respectively” to “respectively (Fig. 5g)” (line 332 in the revised manuscript).

Q40: Lines 344-346: “Furthermore, our... modification on H246”. I think there is something still missing in this mechanism. The direct replacement of the R₂NC(=O)O⁻ group by imidazole is very unlikely from the chemical point of view. The proposed mechanism implies an SN₂ reaction with a poor leaving group. There should be an activation of this group (R₂NC(=O)O⁻) by at least hydrogen bonding with another amino acid from the enzyme. I leave the option of further research on this to the editor.

A: Thank you for the suggestion. In this manuscript, we propose that the nucleophilic attack of the 2'C of the D-ring by the NE2 atom of H246 is mainly based on the Preenzyme calculation and the mass analysis of the **4g**-ShHTL7 reaction system. Michalak et al. (*Tetrahedron* **2014**, *70*, 5073-5081.) reported that after reacting with the nucleophilic reagents, the (*R*)-3-methyl-4-oxocyclopent-2-en-1-yl acetate **R1** could be converted to the (*S*)-4-hydroxy-2-methylcyclopent-2-en-1-one **R2** (Fig. R3). Minai et al. (US4683323A) reported that 4(*S*)-hydroxy-2-cyclopentenone **R3** could be configurationally inverted to the *R*-isomer **R4** (Fig. R3). These suggest that an SN₂ reaction occurs at the 2'C atom upon reacting with nucleophiles.

Fig. R3. SN₂ reactions of the compounds with modified D-ring.

To check whether the direct replacement of the R₂NC(=O)O⁻ group by imidazole is possible or not, we selected compound **4g** as a representative to react with *N*-acetyl-L-histidine methyl ester (Ac-his-ome). The result showed that using NaH as a base and THF as solvent (Supplementary Table 5, entry 1), a new compound with a mass of was 306.1435 was detected by UPLC-HRMS in the reaction solution, which corresponds to the mass of Ac-his-ome-D (HRMS (ESI): calcd for C₁₅H₂₀N₃O₄ [M+H]⁺: 306.1454). Though only a trace amount of the new compound was detected, our results

indicated that the nucleophilic attack on the 2'C atom of **4g** by the imidazole group of histidine is possible. Additionally, we have found that this nucleophilic substitution reaction could also occur under relatively mild reaction conditions (Supplementary Table 5, entries 2 and 3), because we have detected the new compound Ac-his-ome-D in the reaction solution. Agarwal P. K. (*Biochemistry* **2019**, 58, 438–449.) reported that the enzymatic catalytic reaction is far more efficient than the normal chemical reaction. Therefore, it can be inferred that it would be more efficient for the His246 of ShHTL7 to react with **4g** than that under normal conditions, which is also consistent with the results of our experiments. We have added “*In addition, we have found that the nucleophilic attack of the imidazole group of N-acetyl-L-histidine methyl ester to 2'C atom of 4g under the mild reaction conditions is also possible (Supplementary Table 5).*” to the revised manuscript (line 348–350). The new Supplementary Table 5 was added to line 282–286 of the revised Supplementary Information.

Supplementary Table 5. The nucleophilic substitution reaction conditions of Ac-his-ome to the 2'C atom of **4g**.^a

entry	additive (equiv)	solvent	time (h)	temperature (°C)	yield (%) ^b
1	NaH (1.2)	THF	6	0 to 25	trace
2	K ₂ CO ₃ (1.25)	DMF	6	25	trace
3	DBU (1.5)	THF	6	25	trace

^aReactions were performed on a 0.033 mmol scale of **4g** and 10 mL of solvent under the given conditions. ^bThe yields were determined by UPLC-HRMS.

The synthetic methods for compound Ac-his-ome-D are shown below and in line 697–733 of the revised Supplementary Information.

Supplementary Note. Synthesis of compound Ac-his-ome-D.

NaH (60%, 1.6 mg, 0.04 mmol) was added to a solution of Ac-his-ome (7 mg, 0.033 mmol) in tetrahydrofuran (THF, 10 mL) at 0 °C under N₂ with stirring. After 10 min, **4g** (15 mg, 0.033 mmol) was added to the mixture. Next, the reaction was moved to room temperature and stirred for six hours. 0.2 mL of the reaction mixture was taken out by a syringe and diluted with 0.8 mL of acetonitrile,

and the solution was filtrated and used for the UPLC-HRMS analysis. A new compound with a mass of 306.1435 was detected, which corresponded well with the mass of Ac-his-ome-D (HRMS (ESI): calcd for $C_{15}H_{20}N_3O_4$ $[M+H]^+$: 306.1454).

A new compound ($m/z = 306.1435$) corresponding to Ac-his-ome-D in the reaction solution was detected by HPLC-HMRS analysis.

K_2CO_3 (5.7 mg, 0.042 mmol) was added to a solution of Ac-his-ome (7 mg, 0.033 mmol) in *N,N*-dimethylformamide (DMF, 10 mL) at 25 °C with stirring. After 10 min, **4g** (15 mg, 0.033 mmol) was added to the mixture. Next, the reaction was stirred for six hours. 0.2 mL of the reaction mixture was taken out by a syringe and diluted with 0.8 mL of acetonitrile, and the solution was filtrated and used for the UPLC-HRMS analysis. A new compound with a mass of 306.1430 was detected, which corresponded well with the mass of Ac-his-ome-D (HRMS (ESI): calcd for $C_{15}H_{20}N_3O_4$ $[M+H]^+$: 306.1454).

A new compound ($m/z = 306.1430$) corresponding to Ac-his-ome-D in the reaction solution was detected by HPLC-HMRS analysis.

1,8-diazabicyclo[5.4.0]undec-7-ene (DBU, 7.6 mg, 0.05 mmol) was added to a solution of Ac-his-ome (7 mg, 0.033 mmol) in tetrahydrofuran (THF, 10 mL) at 25 °C under. After 10 min, **4g** (15 mg, 0.033 mmol) was added to the mixture. Next, the reaction was moved to room temperature and stirred for six hours. 0.2 mL of the reaction mixture was taken out by a syringe and diluted with 0.8 mL of acetonitrile, and the solution was filtrated and used for the UPLC-HRMS analysis. A new compound with a mass of 306.1436 was detected, which corresponded well with the mass of Ac-his-ome-D (HRMS (ESI): calcd for C₁₅H₂₀N₃O₄ [M+H]⁺: 306.1454).

A new compound ($m/z = 306.1436$) corresponding to Ac-his-ome-D in the reaction solution was detected by HPLC-HMRS analysis.

Yasui et al. (*Angew. Chem. Int. Ed.* **2020**, *59*, 13479–13483.) reported that the chiral phasetransfer catalyst (PTC) could promote an S_N2-type nucleophilic substitution reaction of γ -chlorobutenolide (D-Cl) with *O*-alkylation of enols. They proposed that the intermolecular hydrogen-bonding interactions between the catalyst and the reactants were crucial for reaction. Our molecular docking and the MD simulation studies suggested that 2O and 3O atoms of (*S*)-**4a** could form hydrogen-bonding interactions with S154 and Y174, respectively, these interactions which, in turn, could activate the leaving group (R₂NC(=O)O⁻) and promote the S_N2 reaction. In line 342–344 of the revised manuscript, we have added: “, the hydrogen-bonding interactions of Thr157 and 2O atom, and Tyr174 and 3O atom could active the leaving group (Supplementary Table 4)⁴⁴,”. (ref 44 is Yasui M, Yamada A, Tsukano C, Hamza A, Pápai I, Takemoto Y. Enantioselective acetalization by dynamic kinetic resolution for the synthesis of γ -alkoxybutenolides by thiourea/quaternary ammonium salt

catalysts: application to strigolactones. *Angew. Chem. Int. Ed.* **59**, 13479-13483 (2020).) The new Supplementary Table 4 was added to line 274–275 of the revised Supplementary Information.

Supplementary Table 4. The hydrogen analysis of (*S*)-**4a**–ShHTL7 system in the 1000 ns of the MD.

#Acceptor	DonorH	Donor	AvgDist/Å	AvgAng/°
RES_269@O2	THR_157@HG1	THR_157@OG1	2.90	160.30
RES_269@O3	TYR_174@HH	TYR_174@OH	3.01	143.36
RES_269@O5	MET_96@H	MET_96@N	3.25	135.70
RES_269@O5	TYR_26@H	TYR_26@N	3.28	144.76

Additionally, we have also found that the ShHTL7^{S95A} and ShHTL7^{S95C} mutations could hydrolyze **4g** (Supplementary Fig. 24a). Our LC-MS analysis of **4g**-ShHTL7^{S95A} and **4g**-ShHTL7^{S95C} reaction systems showed that after incubating **4g** with ShHTL7^{S95A}, an increase by 94 Da was detected in the **4g** treated ShHTL7^{S95A} protein (Supplementary Fig. 24e), which corresponding to the modified D-ring of **4g** covalently bound to the ShHTL7^{S95A} protein. Similarly, an increase by 96 Da was detected in **4g** treated ShHTL7^{S95C} protein (Supplementary Fig. 24f), which corresponds to the modified D-ring of **4g** covalently bound to the ShHTL7^{S95C} protein. These results showed that the nucleophilic attack of H246 on the 2'C is possible.

Supplementary Fig. 24. **a**, Relative percentage of hydrolysis of compound **4g** by ShHTL7^{S95A}, ShHTL7^{S95C}, and ShHTL7. **b**, LC-MS analysis of the full-length His-tagged ShHTL7 in native condition. **c**, LC-MS analysis of the full-length His-tagged ShHTL7^{S95A} in native condition, the mass of 34699 Da (reduction of 15 Da compared with that of ShHTL7) is the peak of ShHTL7^{S95A}. **d**, LC-MS analysis of full-length His-tagged ShHTL7^{S95C} in native condition, the mass increment of 18 Da compared with that of ShHTL7 suggesting that ShHTL7^{S95C} is in reduction state. **e**, LC-MS analysis of the **4g**-ShHTL7^{S95A} reaction system in the native condition, an increase of *m/z* by 94 Da indicating that the modified D-ring of **4g** covalently linked to the ShHTL7^{S95A}. **f**, LC-MS analysis of the

4g–ShHTL7^{S95C} reaction system in the native condition, an increase of *m/z* by 96 Da indicating that the modified D-ring of **4g** covalently linked to the ShHTL7^{S95C}.

Q41: Line 358: “higher stability” → “lower reactivity”.

A: We have changed “higher stability” to “lower reactivity” (line 366 in the revised manuscript).

Q42: Line 376: “stereo-selectivity” → “stereoselectivity”.

A: We have changed “stereo-selectivity” to “stereoselectivity” (line 385 in the revised manuscript).

Q43: Line 420: “overnight”, write the actual time (“overnight is very inaccurate).

A: We have changed “overnight” to “for 12 hours” (line 431 in the revised manuscript).

Q44: Line 444: “30 °C, 3d” → “30 °C for 3 days”.

A: We have changed “30 °C, 3d” to “30 °C for 3 days” (line 455 in the revised manuscript).

Q45: Line 463: “30°C” → “30 °C” (space).

A: We have added a space between “30” and “°C” (line 474 in the revised manuscript).

Q46: Line 467: “3” → “three”.

A: We have changed “3” to “three” (line 478 in the revised manuscript).

Q47: Line 468: “12 wells” → “12 well”; “The Netherlands” → “the Netherlands”.

A: We have changed “12 wells” to “12 well” and “The Netherlands” to “the Netherlands” (line 479 in the revised manuscript).

Q48: Line 479 and 484: “25 oC” → “25 °C” (use degree sign, not an “o”).

A: We have changed the “25 °C” to “25 °C” (lines 490 and 495 in the revised manuscript).

Q49: Line 488: “stability” → “reactivity”

A: We have changed the “stability” to “reactivity” (line 499 in the revised manuscript).

Q50: Line 492: “1 oC” → “1 °C” (same as Line 479).

A: We have changed the “1 °C” to “1 °C” (line 503 in the revised manuscript).

Q51: Line 495: “50: 50” → “50:50”.

A: We have deleted the “space” between “50” and “50” (line 506 in the revised manuscript).

Q52: Line 558: “H.B.J.” → “H.J.B.”.

A: We have changed the “H.B.J.” to “H.J.B.” (line 705 in the revised manuscript).

Q53: Lines 736-738: “h, A proposed...”. Needs some addition including some activation of the leaving group.

A: We have added: “*The hydrogen-bonding interactions of Thr157 and 2O atom, and Tyr174 and 3O atom could activate the leaving group,*” (line 763–764 in the revised manuscript).

Supplementary information

Q54: Fig. 1, 2, and 4, see above.

A: We have made the changes in Supplementary Fig. 1, 2, and 4 accordingly.

Q55: Lines 128 and 129: change “stability” accordingly to the text on the article.

A: We have changed “stability” to “reactivity” (line 142–143 in the revised Supplementary information)

Q56: Lines 168-173: redundant information. Only one enantiomer is sufficient.

A: We have removed the ESP results of compound (*R*)-**4a** and only show the ESP results of compound (*S*)-**4a** (line 181–187 in the revised Supplementary information)

Q57: Supplementary Fig. 17: numbers “–0.552” and “–0.551” should be the same. All the - signs (hyphens) should be replaced with – signs (minus signs).

A: We have recalculated the Mulliken atomic charges using different Linux clusters and obtained the same results. According to the above suggestions, we have deleted the Mulliken atomic charges results of (*R*)-**4a** and only show the results of (*S*)-**4a**. Additionally, all the - signs (hyphens) have been replaced with – signs (minus signs) (line 188–194 in the revised Supplementary information).

Q58: Supplementary Fig. 18a: the arrows of the molecule between brackets are wrong. An arrow is missing from the minus charge to form a CO double bond. On ABC-OH, the arrow should come from the nitrogen, not from the CN double bond.

A: We have added an arrow from the O[–] atom to the C–O[–] single bond. Additionally, on the ABC–OH, the arrow has been revised from the nitrogen atom to the carbon atom of the C=O group. (line 195 in the revised Supplementary information).

Q59: Supplementary Table 1: the structure of (3R)-**3g** is wrong. It is supposed to be the enantiomer

of (3S)-3g and it is not.

A: We have redrawn the structure of (2R)-3g to make it correct. (line 262–265 in the revised Supplementary information).

Q60: Line 250: it should include the configurations of the carboxylic acid (many isomers possible).

A: We have changed this sentence as: "...bicyclo[2.2.1]hept-5-ene-2-carboxylic acid (*mixture of endo and exo*)..." (line 300 in the revised Supplementary information)

Q61: Line 253: "overnight", use actual time.

A: We have changed "overnight" to "for 12 hours" (line 303 in the revised Supplementary information)

Q62: Lines 271 and 272: "(d, J = 8.0 Hz, 2H)" → "(m, 2H). This signal is not a doublet, it is part of an AA'BB' system and it is therefore a six-peak multiple (not a doublet).

A: We have changed "(d, J = 8.0 Hz, 2H)" to "(m, 2H) (line 321 in the revised Supplementary information)

Q63: Lines 280 and 281: it should include the configurations of the aldehyde (many isomers possible).

A: We have changed this sentence as: "...bicyclo[2.2.1]hept-5-ene-2- carbaldehyde (*mixture of endo and exo*)..." (line 331 in the revised Supplementary information)

Q64: Lines 325 and 326: Wrong way of reporting the ¹³C spectrum. There are not that many carbon atoms. Fewer carbon atoms should be reported with the coupling constant with fluorine.

A: We have added the C–F coupling constants in the ¹³C spectrum (line 376–378 in the revised Supplementary information)

Q65: Lines 385-387: same as in lines 325, 326.

A: We have added the C–F coupling constants in the ¹³C spectrum (line 437–438 in the revised Supplementary information)

Q66: Line 406: wrong configuration of the molecule. Change.

A: We have redrawn the structure of (2R)-3g to make it correct. (line 457 in the revised Supplementary information).

Q67: Line 477: "(d, J = 8.4 Hz, 2H)" (twice) → "(m, 2H). Same as Lines 271, 272.

A: We have changed "(d, J = 8.4 Hz, 2H)" (twice) to "(m, 2H) (line 528 in the revised

Supplementary information).

Q68: Lines 579-582: same as Lines 325, 326.

A: We have added the C–F coupling constants in the ^{13}C spectrum (line 626–628 in the revised Supplementary information)

Q69: Lines 596-597: same as Lines 325, 326.

A: We have added the C–F coupling constants in the ^{13}C spectrum (lines 641 and 642 in the revised Supplementary information)

Q70: Lines 609-610: same as Lines 325, 326.

A: We have added the C–F coupling constants in the ^{13}C spectrum (line 655–657 in the revised Supplementary information)

Q71: Line 637: “(d, $J = 8.8$ Hz, 2H)” and “(d, $J = 9.2$ Hz, 2H)” → “(m, 2H). Same as Lines 271, 272.

A: We have changed “(d, $J = 8.8$ Hz, 2H)” and “(d, $J = 9.2$ Hz, 2H)” to “(m, 2H)” (line 683 in the revised Supplementary information).

Q72: Line 645: “(d, $J = 7.2$ Hz, 2H)” and “(d, $J = 7.6$ Hz, 2H)” → “(m, 2H). Same as Lines 271, 272.

A: We have changed “(d, $J = 7.2$ Hz, 2H)” and “(d, $J = 7.6$ Hz, 2H)” to “(m, 2H)” (line 690 in the revised Supplementary information).

REVIEWERS' COMMENTS

Reviewer #1 (Remarks to the Author):

The revised manuscript provides much stronger evidence of H246 of ShHTL7 being able to act as a nucleophile to hydrolyze compounds with a modified butenolide ring, particularly the LC-MS showing presence of a butenolide modification in the S95A and S95C mutants. One final suggestion is for the authors to comment on whether a strigolactone compound with an unmodified butenolide ring (oxygen at the 1' position) would be more likely to undergo this new SN2-like pathway (H246-nucleophile) or the S95-nucleophile pathway shown in Fig. S19A. However, this revised manuscript is much improved over the original, and I recommend it for publication pending this comment.

Reviewer #2 (Remarks to the Author):

The authors have addressed all my concerns. I have no further comments.

Reviewer #3 (Remarks to the Author):

This publication presents very relevant results in the field of crop protection against some parasitic weeds like witchweeds and broomrapes. There's been a lot of research on this field in the last decades looking for active germinating stimulants that can be used for the suicidal germination approach to try to save billions of dollars a year.

Xi et al. present a series of potential compound candidates (small-molecule agonists that stimulate *Phelipanche aegyptiaca* and *Striga hermonthica*) for solving this problem with an activity comparable (or even better) than the one of GR24 (the reference compound in the field). The authors also present their study and advances on the elucidation of the mechanism of action of these compounds, which could accelerate the progress on the synthesis of new active molecules to eliminate the parasitic weeds. Their research showed that full efficiency of the synthetic SL agonists in triggering signaling through the *Striga* SL receptor (ShHTL7) depends on the receptor catalyzed hydrolytic reaction of the agonists. They also proposed a new hydrolytic mechanism, although it should be considered just a proposal because more evidence is required to prove it.

The work presented in this publication supports efficiently most of the conclusions and claims drawn at the end of the study and meets the expected standards in the field. All the results are presented with enough detail to be reproduced by an expert in the field.

I proposed making major revisions on the article (see doc file) before it could be published.

After having received the revised version, I can say that I am very satisfied with the work done for the revision of the manuscript.

Daniel Blanco-Ania

Point-by-point responses to Reviewers' Comments

We thank all three reviewers once more for their valuable time and effort throughout the review process to help us improve the manuscript. We have responded to the comment raised by reviewer 1 as indicated below.

REVIEWERS' COMMENTS

Reviewer #1 (Remarks to the Author):

The revised manuscript provides much stronger evidence of H246 of ShHTL7 being able to act as a nucleophile to hydrolyze compounds with a modified butenolide ring, particularly the LC-MS showing presence of a butenolide modification in the S95A and S95C mutants. One final suggestion is for the authors comment on whether a strigolactone compound with an unmodified butenolide ring (oxygen at the 1' position) would be more likely to undergo this new SN2-like pathway (H246-nucleophile) or the S95-nucleophile pathway shown in Fig. S19A. However, this revised manuscript is much improved over the original, and I recommend it for publication pending this comment.

Response: We thank the reviewer for the approval for our publication on NC. We really appreciate your suggestion on the possibility of the reaction between strigolactone compound with an unmodified butenolide ring (oxygen at the 1' position) and SL receptor in the natural signaling process. So far, as illustrated in Supplementary Fig. 11, the hydrolysis percentage by ShHTL7 was decreased when the 1'O of the D-ring of compound **4a** was changed to methylene (**4g**) and *N*-CH₃ (**4h**) groups, suggesting that the hydrolysis rate of the SN2-like pathway is slower than that of S95-nucleophile pathway. So, in the current manuscript, we still inferred that for the SL analog containing an unmodified butenolide ring (oxygen at the 1' position), the hydrolysis would mainly undergo the S95-nucleophile pathway. Of course, we certainly realized that the possibility indicated by this referee might imply a new signaling pathway if this can really happen in nature; we are currently pursuing this research. Theoretically, ShHTL7 may generate this possible reaction and make it a little bit favored, but we still have no solid evidence on this. We hope we can report our result on this aspect in due course.

Reviewer #2 (Remarks to the Author):

The authors have addressed all my concerns. I have no further comments.

We thank Reviewer #2 for the review and improvement of our manuscript.

Reviewer #3 (Remarks to the Author):

This publication presents very relevant results in the field of crop protection against some parasitic

weeds like witchweeds and broomrapes. There's been a lot of research on this field on the last decades looking for active germinating stimulants that can be used for the suicidal germination approach to try to save billions of dollars a year.

Xi et al. present a series of potential compound candidates (small-molecule agonists that stimulate *Phelipanche aegyptiaca* and *Striga hermonthica*) for solving this problem with an activity comparable (or even better) than the one of GR24 (the reference compound in the field). The authors also present their study and advances on the elucidation of the mechanism of action of these compounds, which could accelerate the progress on the synthesis of new active molecules to eliminate the parasitic weeds. Their research showed that full efficiency of the synthetic SL agonists in triggering signaling through the Striga SL receptor (ShHTL7) depends on the receptor catalyzed hydrolytic reaction of the agonists. They also proposed a new hydrolytic mechanism, although it should be considered just a proposal because more evidence is required to prove it.

The work presented in this publication supports efficiently most of the conclusions and claims drawn at the end of the study and meets the expected standards in the field. All the results are presented with enough detail to be reproduced by an expert in the field.

I proposed making major revisions on the article (see doc file) before it could be published.

After having received the revised version, I can say that I am very satisfied with the work done for the revision of the manuscript.

Daniel Blanco-Ania

Thank you, Dr. Blanco-Ania, for the constructive comments and recommendation.